# AdaVideoRAG:
# Omni-Contextual Adaptive Retrieval-Augmented Efficient Long Video Understanding

**Zhucun Xue**[1*]   **Jiangning Zhang**[1,2*] **Xurong Xie**[1]   **Yuxuan Cai**[3]   **Yong Liu**[1†] **Xiangtai Li**[4]
**Dacheng Tao**[4]

[1]Zhejiang University   [2]Youtu Lab, Tencent   [3]Huazhong University of Science and Technology
[4]Nanyang Technological University
Code: https://github.com/xzc-zju/AdaVideoRAG

## Abstract

Multimodal Large Language Models (MLLMs) have demonstrated excellent performance in video understanding but suffer from degraded effectiveness when processing long videos due to fixed-length contexts and weaknesses in modeling long-term dependencies. Retrieval-Augmented Generation (RAG) technology can mitigate these limitations through dynamic knowledge expansion, but existing RAG schemes for video understanding employ fixed retrieval paradigms that use uniform structures regardless of input query difficulty. This introduces redundant computational overhead and latency (*e.g.*, complex graph traversal operations) for simple queries (*e.g.*, frame-level object recognition) while potentially causing critical information loss due to insufficient retrieval granularity for multi-hop reasoning. Such single-step retrieval mechanisms severely constrain the model's balance between resource efficiency and cognitive depth. To address this, we first propose a novel AdaVideoRAG framework for long-video understanding, which uses a lightweight intent classifier to dynamically and adaptively allocate appropriate retrieval schemes, ranging from the simplest to the most sophisticated, for different video understanding tasks based on query complexity. We introduce an Omni-Knowledge Indexing module to extract valuable information from multi-modal signals for context modeling and build corresponding databases, *i.e.*, a text base from clip captions, ASR, and OCR; a visual base; and a graph for deep semantic understanding. This enables hierarchical knowledge access, integration, and generation from naive retrieval to graph retrieval, achieving an optimal balance between resource consumption and video understanding capabilities. Finally, we construct the HiVU benchmark for deep understanding evaluation. Extensive experiments show that our framework enhances the overall efficiency and accuracy of Video-QA for long videos and can be seamlessly integrated with existing MLLMs via lightweight API calls, establishing a new paradigm for adaptive retrieval augmentation in video analysis.

## 1 Introduction

With its powerful multimodal perception and generalization capabilities, the Multimodal Large Language Model (MLLM) has become a universal technical paradigm for addressing diverse scenarios and has demonstrated strong generative capabilities in video understanding [31, 48, 29, 1]. However,

---

[*]Equal contributions.
[†]Corresponding author.

39th Conference on Neural Information Processing Systems (NeurIPS 2025).

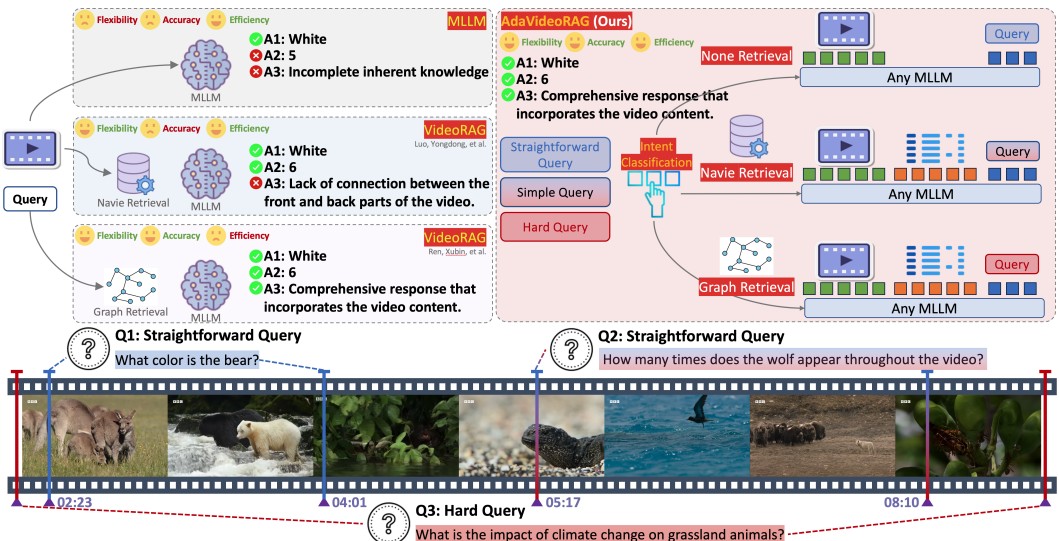

Figure 1: Comparison of different video understanding frameworks: *i)* MLLMs are efficient but can only handle simple problems. *ii)* VideoRAG [32] integrates external knowledge via naive retrieval but still struggles with hard reasoning questions. *iii)* Recent VideoRAG [36] tackles complex problems using graph retrieval but suffers from low efficiency. Our novel AdaVideoRAG framework adaptively routes queries to different retrieval paths via query intent classification, achieving a better trade-off between effectiveness and efficiency.

when applied to specific domains, it is constrained by challenges such as **knowledge solidification** (inability to dynamically update the latest knowledge), **uncontrollable reasoning** (risk of hallucinations), and **weak generalization** (requiring additional fine-tuning costs and time costs), making it difficult to handle multi-hop question and cross-modal association requirements (especially in long video scenarios), which leads to performance degradation [36, 32]. Retrieval-Augmented Generation (RAG), by integrating the collaborative reasoning of external knowledge bases and generative models without being confined to pre-trained knowledge, can easily adapt to private domain data scenarios and has become a core paradigm for improving the factual accuracy and domain adaptability of large language models.

Current RAG research mostly focuses on text modality [25, 10, 19], static images [7], and tabular forms [6], overlooking the unique value of video as a multimodal knowledge carrier. The increasingly popular long-video understanding has put forward new demands for RAG models supporting video modality input. Most existing RAG studies on long videos attempt to enhance question-answering generation by constructing and retrieving knowledge bases from multimodal information derived from videos. For example, Luo *et al.* [32] incorporates visually-aligned auxiliary text features from optical character recognition (OCR), automatic speech recognition (ASR), and object detection to create video knowledge bases, enabling question-answering for long videos. However, this method does not support sensemaking queries or multi-hop questions, which require global understanding of the entire database as shown in Fig. 1. Recent VideoRAG [36] significantly improves the accuracy of long-video contextual information by constructing a graph database, but it requires maintaining a hierarchical graph database that demands substantial computational and time resources, and incurs higher costs when migrating to new scenarios. We believe that a practical RAG for video understanding needs to flexibly allocate appropriate processing methods for different videos and query difficulties, which both maintains accuracy and improves efficiency.

Considering that real-world video understanding tasks involve content comprehension needs of varying complexity, the problem-solving strategies for questions of different difficulty levels will have distinct priorities. Short-video QA involving simple common sense does not require retrieval and can directly obtain correct answers by querying the MLLM, while complex long-video questions rely on RAG for retrieval to filter effective information. For more complex questions, such as those requiring multi-step reasoning or relying on multiple types of knowledge graph based RAG is necessary to derive correct answers. Therefore, a one-size-fits-all approach of retrieving and then returning results

is not optimal. To address this, this paper proposes an adaptive-RAG-based video understanding scheme termed AdaVideoRAG, as shown in Fig. 1. It first classifies user queries into difficulty levels and then adaptively assigns the most reasonable retrieval strategy based on the difficulty. Additionally, we further integrate visual features, clip captions, ASR, and scene text composite information flows contained in videos, and use relevant text information obtained from external retrieval for data augmentation. According to the difficulty of questions, queries are routed to different levels of database retrieval modes (*i.e.*, naive and graph retrieval). These multimodal knowledge inputs and retrieval strategies can more effectively provide fine-grained contextual representation capabilities, ultimately further enhancing the upper limit of MLLM's processing capabilities for long videos and complex question-answering tasks.

To demonstrate the effectiveness of the proposed AdaVideoRAG framework, we officially release HiVU, the first open benchmark dataset for full-stack capability evaluation in video understanding. This dataset groundbreakingly integrates 120 video samples covering a continuous duration spectrum from short clips (1 minute) to extra-long videos (106 minutes), spanning high-frequency scene categories across three major themes: knowledge education (lectures, finance, law, psychology, documentaries), information (news, interviews), and entertainment (sports, cooking, makeup, fitness, TV dramas, animations). In terms of question design, we innovatively develop a three-level difficulty quantification system: *1)* **Basic Level-1 (L1)** focuses on frame-level content perception (e.g., "Which objects appear at the 5th second of the video?"). *2)* **Advanced Level-2 (L2)** requires temporal logic reasoning (e.g., "When does the speaker start explaining graph neural networks?"). *3)* **Expert Level-3 (L3)** challenges cross-modal causal inference (e.g., "How would deleting the narration at the 15th minute affect the plot development?"). Compared with traditional datasets such as ActivityNet [2] (single action recognition) and MovieQA [37] (open-ended QA), this benchmark achieves, for the first time, cognitive complexity evaluation at different levels, providing a hierarchical evaluation framework for video understanding research. It supports systematic optimization of models in long-video modeling, complex reasoning tasks, and real-world scenario generalization. In summary, our contributions are as follows:

*1)* We propose a novel AdaVideoRAG framework that dynamically and adaptively routes appropriate retrieval schemes, from the simplest to the most sophisticated, depending on the query complexity. This approach aims to achieve an optimal balance between resource consumption and video understanding capabilities for different video understanding tasks.

*2)* We introduce an Omni-Knowledge Indexing module to extract valuable information from multi-modal signals for context modeling and establish corresponding databases. A lightweight intent classification model is used to determine the difficulty level of input queries, enabling hierarchical knowledge access, integration, and generation from naive retrieval to graph retrieval, while balancing resource consumption and video understanding capabilities.

*3)* We publicly release the hierarchical video understanding benchmark HiVU for the first time, which evaluates the multi-level reasoning capabilities of video understanding models. Extensive comparative experiments and ablation studies demonstrate the advantages of AdaVideoRAG in deep understanding of long videos.

## 2 Method

We introduce an MLLM-centric adaptive RAG framework for long-video understanding termed AdaVideoRAG, which can significantly improve efficiency while ensuring accuracy. As shown in Fig. 2, our method includes four parts: 1) Query Intent Classification (Sec. 2.1). 2) Omni-Knowledge Indexing (Sec. 2.2). 3) Adaptive Retrieval Paradigm (Sec. 2.3). 4) Integration and Generation (Sec. 2.4).

### 2.1 Query Intent Classification

Not all user requests have the same level of complexity. For simple user requests, we can use a straightforward solution to reduce computing power consumption and users' perception of latency. For complex questions, we rely on complex multi-model, multi-modal, and multi-step queries to achieve higher accuracy. To achieve the above goals, we propose to use a lightweight intent classification model to perform the classification of the difficulty level of the query at the input end. Specifically, we have defined and established a fine-grained evaluation system for the difficulty level of video understanding:

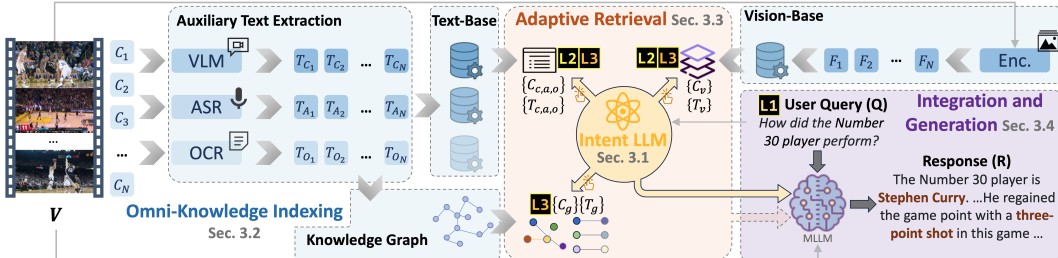

Figure 2: **Overview of our AdaVideoRAG framework** that consists of: *1)* Query Intent Classification (Sec. 2.1). *2)* Omni-Knowledge Indexing (Sec. 2.2). *3)* Adaptive Retrieval Paradigm (Sec. 2.3). *4)* Integration and Generation (Sec. 2.4).

**Level-1: Straightforward reasoning.** There are basically few logical relationships involved in the questions, and the knowledge required for answering questions is directly provided in the video content. For example, "What color of clothes is the woman who appears at the fifth second wearing?" For such questions, the existing MLLMs models are already very mature in solving them. If complex processing is still applied to such simple queries, it will result in unnecessary computational overhead.

**Level-2: Simple reasoning.** It involves single-step reasoning about basic spatio-temporal/causal relationships, requiring the model to establish logical associations between local events. For example, "Why did the woman cry before the rainy scene started?" requires two-stage reasoning: 1) Determine the starting time point of "rain" through temporal positioning; 2) Retrieve the character behaviors (such as the audio of an argumentative conversation) and scene changes (such as weather forecast subtitles) before this time point, and construct a causal chain to explain the motivation. Such tasks expose the integration flaws of existing MLLMS methods regarding cross-modal temporal clues, and are prone to the lack of key intermediate evidence due to the mismatch in retrieval granularity.

**Level-3: Hard reasoning.** The video understanding at the highest difficulty level requires extracting different subjects and relationships from the long-context, and constructing a knowledge graph that maps entities and relationships across temporal and semantic dimensions, and combining it with powerful MLLM reasoning capabilities to make judgments. For example, "What life lessons does this movie convey?" Questions of this kind require the model to mine the deep semantic relationships provided in the video and conduct multi-hop reasoning to obtain the correct answers.

**Intent classification model.** Given the basic definitions and examples from level 1 to level 3, we use a large language model with Chain-of-Thought (CoT) reasoning to classify the query $Q$. This can be integrated into a RAG (Retrieval-Augmented Generation) architecture as a plug-and-play API, providing intent classification results through appropriate prompts without the need for fine-tuning. Based on the classification results, it can automatically trigger a progressive knowledge retrieval strategy, ranging from none retrieval to simple naive retrieval, and further to complex graph retrieval. The calculation of the intent classification result $L$ can be formulated as: $L = LLM_{intent}(Q, prompt_{intent})$. where the LLM is a lightweight CoT model. In this paper, we adopt Qwen2.5-7B [18, 45], whose time-consuming proportion is extremely small (averagely $\leq 5\%$) compared to the entire process.

## 2.2 Omni-Knowledge Indexing for Long-Context Understanding

When performing video understanding tasks, MLLMs equipped with RAG can achieve context modeling through dynamic activation of external knowledge bases, which alleviates the window length limitation of long contexts to some extent and enhances the semantic understanding of global videos. To this end, we propose the Omni-Knowledge Indexing module, which extracts valuable information from multiple modal signals for context modeling and establishes corresponding databases, enabling the RAG system to more accurately retrieve the most relevant information and perform high-quality generation.

### 2.2.1 Omni-Knowledge Text-Base Establishment

In long video understanding tasks, due to the context window size limitations of MLLMs, we need to perform frame sampling and resizing on videos under hardware constraints. However, this inevitably leads to the loss of rich visual information in the videos, as well as unused audio and text multimodal information. Therefore, we utilize an external normalization module to extract multimodal information from videos and construct our private text base.

**Auxiliary text extraction and database construction.** The input long video $V$ is divided into $N$ consecutive and semantically complete clips $V = (C_1, C_2, \ldots, C_N) = \{C_n\}$ at fixed time intervals (30 seconds per clip in the paper). For each clip $C_n$, uniform frame sampling is performed to extract key frames. In this paper, we select 5 frames as the multimodal representation primitive $F_n$, as more frames do not significantly improve performance but increase computational power and model complexity. Specifically, auxiliary text extraction includes three categories: *1)* The quantized MiniCPM-V [46] (used as the VLM model) generates fine-grained text descriptions $T_C$ for the sampled frames, including semantic elements such as character attributes and spatio-temporal actions, ultimately constructing a caption database $D_C$; *2)* Audio is the most direct information carrier in videos, driving story development, conveying plot clues, and revealing character relationships through language, providing information that cannot be mined from visual features alone. Therefore, we use FastWhisper [33] as the audio extractor to convert the audio in each clip into text format $T_A$, which is stored as vectors via an embedding model to generate an ASR database $D_A$; *3)* Characters $T_O$ in each frame are extracted through EASYOCR [22], and an OCR database $D_O$ is constructed to compensate for the insufficient recognition ability of MLLMs.

**Knowledge graph construction.** To address Level-3 complex reasoning queries, we construct a knowledge graph based on clip captions ($T_C$), ASR ($T_A$), and OCR ($T_O$). Specifically, BGE-M3 [5] extracts entities and relationships from text chunks: *1) **Entity*** represents the minimal domain-specific semantic interpretation unit in the video, characterized by a triple <entity type, entity name, spatio-temporal attribute>. *2) **Relationship*** encompasses various semantic associations between entities, including spatio-temporal relationships, causal relationships, functional relationships, etc., to systematically structure video text information.

### 2.2.2 Vision-Base Establishment

Simply relying on text information extracted from clip captions, ASR, and OCR makes it difficult to construct an optimal Knowledge Indexing. As a typical carrier of multimodal data, videos contain visual features with abundant details that are hard to describe precisely in text, such as object appearance changes, scene spatial layouts, and human facial expressions and movements. These visual information play an indispensable role in complex knowledge reasoning and retrieval tasks. Therefore, we introduce the ImageBind [16] image encoder (Enc. in Fig. 2) to extract features from key frames and concatenate them as the final features, because this model is based on advanced cross-modal alignment algorithms that can map heterogeneous modal data such as images, text, and audio into the same high-dimensional semantic space.

## 2.3 Adaptive Retrieval Paradigm

After intent classification (Sec. 2.1), the user query (Q) is routed to different retrieval paths according to its difficulty level, so as to improve comprehensive efficiency on the premise of ensuring effectiveness.

**None retrieval with direct MLLM.** For Level-1 scenarios, the model directly feeds the query (Q) and the entire video $\{C_n\}$ into the MLLM to obtain a direct response. This approach leverages the inherent knowledge and reasoning capabilities of the MLLM without introducing external knowledge bases, significantly enhancing overall efficiency for simple questions.

**Naive retrieval with simple reasoning.** For Level-2 retrieval scenarios, this study proposes a multimodal collaborative grounding framework that significantly enhances the retrieval efficiency and accuracy of long videos in handling simple logical questions by jointly optimizing the semantic alignment between auxiliary texts (clip captions, ASR, OCR) and visual modalities. Specifically, we first decouple the original query into sub-queries adapted to different modal characteristics: *1)* For clip caption retrieval, we rewrite the query into declarative sentences, remove option interference, and add scene-appropriate descriptive content. *2)* For ASR-recognized text, we extract colloquial

expressions from the query, retain core actions and events, and add contextual modifiers to match fragmented speech segments. *3)* For discrete OCR text, we extract specific entity information from the query. A typical example: when the input query is *"How did the Number 30 player perform?"*, the rewritten outputs are: *i)* "clip caption": *"The performance of Number 30 player."*; *ii)* "ASR text": *"How's the number 30 player doing."*; *iii)* "OCR text": *"Number 30 player"*. Query rewriting effectively mitigates distribution shifts between different semantics. Through cross-modal similarity calculation, we can then quickly locate query-relevant candidate content and the corresponding video clips for each text block.

This study further locates and queries the semantically most relevant video content from the visual feature database $D_V$. Specifically, our model reuses the rewritten results of clip captions as semantic anchors for visual retrieval. The pre-trained cross-modal semantic alignment encoder ImageBind [16] is employed to map videos into the text embedding space $\{F_n\}$. By calculating the cosine similarity between text and visual embeddings, candidate segments with similarity scores exceeding a threshold (set to 0.5 in this paper) are filtered out. These segments are then ranked to retain the top-K visual evidence with the highest confidence. This approach significantly reduces the modality gap in visual-text alignment by leveraging a unified semantic embedding space, effectively alleviating the problem of local detail loss in long videos. Finally, the videos $\{C_v\}$ retrieved through visual feature-text space alignment are merged with the video chunks $\{C_{c,a,o}\}$ located via auxiliary text retrieval to construct a retrieval evidence pool for simple reasoning at Level-2.

**Graph retrieval in hard reasoning.** Relying solely on information obtained from auxiliary text and visual feature retrieval falls short in enabling MLLMs to tackle more complex sensemaking query scenarios. Therefore, we require more abundant and semantically precise auxiliary information capable of modeling multiple events and temporal nodes to facilitate MLLM reasoning. To address this challenge, we adopt a deeper retrieval approach based on Light-RAG [19] to handle hard queries, replacing the naive retrieval method used for simple queries. Specifically, considering resource constraints, we reuse auxiliary text embeddings to construct a graph. We then compute similarity scores between rewritten clip captions and entity/relationship descriptions, returning the most relevant entities and relationships. Within the graph map, we gather other information associated with the retrieved entities and relationships, which can be combined into a query-centered thinking map. This retrieved graph map assists MLLMs in considering global and multi-layered information, aiding in better modeling spatio-temporal and causal relationships within events. Furthermore, we employ a unified semantic embedding space to represent visual evidence obtained from grounding, enhancing retrieval accuracy. We overlay the retrieved videos $\{C_v\}$ with graph retrieval results $\{C_g\}$ to construct a multi-level retrieval evidence pool for hard reasoning under Level-3.

**Filtering then sorting evidences.** After obtaining the preliminary retrieval results, we perform coarse-to-fine information purification on the search results. First, we filter out duplicate video information blocks retrieved from different databases. Then, the content description of the video blocks (including clip captions, ASR, and OCR texts) and the query are simultaneously input into a small-scale LLM (Qwen2.5-7B [45, 18] in the paper) for fine-grained filtering to exclude some irrelevant search results. Finally, we rerank the selected video clips based on the order of original video time to preserve temporal causal relationship information.

## 2.4 Multimodal Information Integration and Generation

To provide MLLMs with more comprehensive information for enhancing query accuracy, we acquire auxiliary text information (denoted as $\{T_{c,a,o}\}$ for simple reasoning and $\{T_g\}$ for hard reasoning) derived from clip captions, ASR, and OCR contexts, along with visual information $\{C_v\}$ from visual-to-text grounding. After integrating the retrieved context and corresponding video clips $\{C_{c,a,o}\}$, the combined inputs are fed into MLLMs for reasoning and generation to produce the final output $R$:

$$R = \begin{cases} \text{MLLM}(\{C_n\}, Q) & \text{if } L \text{ is Level-1,} \\ \text{MLLM}(\{C_v\}, \{C_{c,a,o}\}, \{T_{c,a,o}\}, Q) & \text{if } L \text{ is Level-2,} \\ \text{MLLM}(\{C_v\}, \{C_{c,a,o}\}, \{T_{c,a,o}\}, \{C_g\}, \{T_g\}, Q) & \text{if } L \text{ is Level-3.} \end{cases} \tag{1}$$

## 2.5 HiVU: Hierarchical Video Understanding Benchmark

Existing video understanding datasets either have insufficient duration [14] or lack engaging content [43, 53], failing to generate queries that require deep comprehension. To support robust reasoning

tasks on long videos and evaluate different methods, we constructed the Hierarchical Video Understanding (HiVU) Benchmark. For this purpose, we selected three genres: knowledge-education (lectures, finance, law, psychology, documentaries), information (news, interviews), and entertainment (sports, cooking, makeup, fitness, TV dramas, animations). We manually collected 120 long-video datasets rich in knowledge content from YouTube, totaling 60 hours, with distributions shown in Fig. 3. Additionally, we designed three tiers of query reasoning from straightforward to hard, as described in Sec. 2.1. This hierarchical query design enables comprehensive and detailed evaluation of models' reasoning capabilities across varying difficulty levels.

**Evaluation metrics.** For the open-ended question-answering tasks on the HiVU dataset, we draw inspiration from the Win-Rate metric system widely used in the RAG field to evaluate model capabilities [10, 19]. Specifically, we use large language models (LLMs) as the judgment basis, quantify the comparative results of the two schemes through model outputs, and finally present their competitive scores in percentage form. The Win-Rate Comparison comprehensively considers queries from five dimensions: *1)* **Comprehensiveness:** This dimension focuses on whether the model's response fully covers the query, avoiding missing critical information or providing

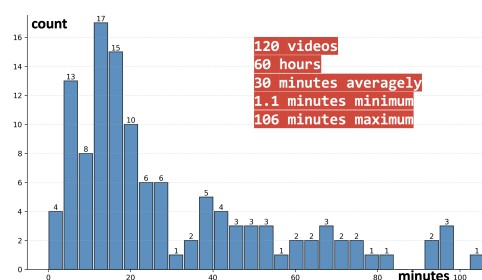

Figure 3: Statistical distributions of our **HiVU** from different perspectives.

one-sided answers. *2)* **Empowerment:** It primarily examines whether the model's response can provide practical value and inspiration to users. *3)* **Trustworthiness:** This dimension emphasizes the reliability and authenticity of the model's output content. *4)* **Depth:** It assesses whether the model can go beyond surface phenomena, uncover the essential issues behind the query, and conduct in-depth analysis and discussion. *5)* **Density:** It focuses on the information content and compactness of the model's response, avoiding verbose, empty, or redundant expressions.

## 3 Experiments

### 3.1 Experimental Setup

We conduct comprehensive evaluations of the proposed AdaVideoRAG method and the effectiveness of each module primarily on the newly proposed HiVU benchmark Sec. 2.5, and also introduce public video understanding benchmarks for further thorough assessment. Specifically: *1)* HiVU includes over 10 sub-genres across 3 domains, comprising 120 knowledge-rich long-video datasets totaling 60 hours. *2)* Video-MME [14] is a full-spectrum multi-modal evaluation benchmark for MLLMs in video analysis, featuring diverse videos and multi-modal data. It contains 900 videos (ranging from 11 seconds to 1 hour, categorized into short, medium, and long), with 2,700 multiple-choice questions covering 6 major visual domains (e.g., knowledge, film, sports) and 30 subdomains, focusing on evaluating the perception, reasoning, and summarization capabilities of multimodal large language models (MLLMs) in video analysis. *3)* MLVU [53] is a multi-task benchmark for evaluating long-video understanding with diverse genres and extended durations. Centered on long videos ranging from 3 minutes to over 2 hours (average 12 minutes), it sets 9 multi-tasks (e.g., single/multi-detail understanding) across diverse video types (films, surveillance, games, etc.), aiming to comprehensively assess long-video understanding capabilities.

### 3.2 Experimental Results

**Improving open-sourced MLLMs with AdaVideoRAG on MLVU_test [53] benchmark.** The overall evaluation results of all the investigated multi-modal large language models in the MLVU test set are shown in Tab. 1. These results cover the baseline model, Video-LLaVA [29], along with two highly regarded open-source models released recently: Qwen2.5-VL series [1] and VideoL-LaMA3 [47]. The evaluation results clearly demonstrate that the AdaVideoRAG strategy we proposed significantly improves the question-answering accuracy of each MLLM. And it particularly stands out in two key types of tasks. Firstly, in tasks such as Topic Reasoning (TR) that require multi-hop reasoning about videos, and secondly, in tasks like Action Count (AC) that involve holistic reasoning. This indicates that AdaVideoRAG can not only strengthen the basic question-answering ability but also

effectively assist the MLLMs in achieving breakthroughs within complex reasoning and multi-detail processing tasks. It is worth noting that although the Qwen2.5-VL-7B model that performs relatively weakly on the MLVU dataset, it exhibits more pronounced accuracy improvements after adopting our AdaVideoRAG, increasing nearly by 40% and even reaching the accuracy of large-parameter models like Qwen2.5-VL-32B. What's more, the open-source model VideoLLaMA equipped with AdaVideoRAG, even though it has fewer parameters than Qwen2.5-VL-32B, shows better performance on long videos, and its performance can even be comparable to that of GPT-4o. These experimental results fully verify the universality and effectiveness of AdaVideoRAG in enhancing the reasoning ability of MLLMs.

Table 1: Comparison between supervised baselines and whether AdaVideoRAG is configured on MLVU_test. **Frames:** the sampling frame rate or the number of images limited, and "2fps-768" indicates that videos are sampled at 2 fps and the upper limit is 768 frames; **M-Avg**: the average performance of multiple-choice tasks.

| Model | Params | Frames | TR | AR | NQA | ER | PQA | SQA | AO | AC | TQA | AVG | Gain |
|---|---|---|---|---|---|---|---|---|---|---|---|---|---|
| GPT-4o | - | 0.5fps | 83.7 | 68.8 | 42.9 | 47.8 | 57.1 | 63.6 | 46.2 | 35 | 48.7 | 54.9 | - |
| Video-LLaVA | 7B | 8 | 64.4 | 35.9 | 25.4 | 34 | 26 | 25 | 13.1 | 16.9 | 23.8 | 29.4 | - |
| Video-LLaVA + AdaVideoRAG | 7B | 8 | 73.9 | 33.1 | 46.2 | 38 | 41.9 | 31.3 | 21.2 | 16.9 | 38.5 | 37.9 | 28.9% |
| Qwen2.5-VL | 7B | 2fps-768 | 46.7 | 15.4 | 16.9 | 35.8 | 38 | 38.9 | 24.6 | 13.6 | 31 | 29.0 | |
| Qwen2.5-VL + AdaVideoRAG | 7B | 2fps-768 | 78.9 | 30.8 | 44.1 | 37.7 | 48 | 36.1 | 33.3 | 15.3 | 40.5 | 40.5 | 39.8% |
| Qwen2.5-VL | 72B | 2fps-768 | 73.3 | 33.3 | 59.3 | 47.2 | 40 | 41.7 | 37.7 | 16.9 | 26.2 | 41.7 | |
| Qwen2.5-VL + AdaVideoRAG | 72B | 2fps-768 | 82.2 | 41 | 54.2 | 41.5 | 44 | 47.2 | 35.1 | 15.1 | 45.2 | 45.1 | 8% |
| VideoLLaMA3 | 7B | 1fps-180 | 76.9 | 43.6 | 68.3 | 54.7 | 58 | 34.3 | 25 | 33.3 | 34.9 | 47.7 | |
| VideoLLaMA3 + AdaVideoRAG | 7B | 1fps-180 | **83.8** | **47.1** | **69.2** | **62.3** | **64** | 38.9 | 34.8 | **35.6** | 42.9 | 53.2 | 11.6% |

**Comparison with state-of-the-art VideoRAG [32] on Video-MME [14] dataset.** Given that the experimental results in Tab. 1 have fully verified that AdaVideoRAG can effectively enhance the reasoning performance of MLLMs, we select the VideoLLaMA3 and the Qwen2.5-VL-7B model as the basic model for subsequent control experiments, which with the same number of parameters. In Tab. 2, we conduct a horizontal comparative test between our AdaVideoRAG and Video-RAG [32] on the Video-MME dataset. The experimental results show that both RAG methods can significantly enhance the video understanding ability of the base MLLMs. However, in tasks involving the processing of long videos, our AdaVideoRAG demonstrates a more distinct advantage. This is mainly due to the fact that AdaVideoRAG is capable of constructing a more complex and reasonable knowledge map during the retrieval of long videos, thus enabling precise understanding and efficient reasoning of video.

Table 2: Comparison between AdaVideoRAG and VideoRAG [32] on Video-MME [14] dataset.

| Model | Params | Frames | Short | Medium | Long | Overall | Gain |
|---|---|---|---|---|---|---|---|
| GPT-4o | - | 384 | 80 | **70.3** | **65.3** | **71.9** | |
| Qwen2.5-VL | 7B | 2fps-768 | 55.6 | 47.1 | 38.8 | 47.2 | |
| Qwen2.5-VL + VideoRAG [32] | 7B | 32 | 70.3 | 51.5 | 43.3 | 55.0 | +7.9 |
| Qwen2.5-VL + AdaVideoRAG | 7B | 2fps-768 | 72.8 | 59.1 | 47.7 | 59.9 | +12.7 |
| VideoLLaMA3 | 7B | 1fps-180 | 76.7 | 62.8 | 53.2 | 64.2 | |
| VideoLLaMA3 + VideoRAG [32] | 7B | 32 | 81.5 | 63.3 | 57.1 | 67.3 | 3.1 |
| VideoLLaMA3 + AdaVideoRAG | 7B | 1fps-180 | **80.3** | 65.4 | 59.8 | 68.5 | 4.3 |

**Impact of LLM arbiters.** To explore the performance of the retrieval strategies in sensemaking tasks of varying difficulties, we conducts comparative experiments based on the proposed hierarchical video understanding benchmark(HiVU), and an LLM is then used as the evaluation referee to assess the quality of the final answers. Regarding the selection of specific LLMs, we carry out two sets of control experiments: Deepseek-R1-7B [45, 18] and Deepseek-R1-32B [45, 18], Qwen2.5-32B [45] and QwQ-32B [39], which represent the models with different parameters and reasoning capabilities respectively, as illustrated in Tab. 3. The experimental results demonstrate that models with larger parameters and equipped with the Chain-of-Thought (CoT) reasoning mechanism exhibit stronger discriminatory abilities when evaluating the performance of other models. Based on these findings, we choose DeepSeek-32B model as the evaluation arbiter for HiVU benchmark evaluation to ensure the accuracy and reliability of the evaluation results.

**Comparison with state-of-the-art VideoRAG [32] on HiVU dataset.** In our HiVU data benchmark, there are three tasks classified according to the difficulty of reasoning: straightforward (L1), simple (L2), and hard (L3). For different levels, AdaVideoRAG employs different retrieval strategies: from without retrieval, naive retrieval, to graph retrieval, which forms a hierarchical enhancement mechanism. And the following series of experiments are designed to verify the improvement of

Table 3: Impact of LLM arbiter configurations (parameter scale and reasoning capabilities) on HiVU benchmark evaluation.

| Metric | Deepseek-7B | | Deepseek-32B | | Qwen2.5-32B | | QwQ-32B | |
|---|---|---|---|---|---|---|---|---|
| | VideoLLaMA3 | VideoLLaMA3 w/ AdaVideoRAG | VideoLLaMA3 | VideoLLaMA3 w/ AdaVideoRAG | VideoLLaMA3 | VideoLLaMA3 w/ AdaVideoRAG | VideoLLaMA3 | VideoLLaMA3 w/ AdaVideoRAG |
| Comprehensiveness | 46.1% | 53.9% | 35.98% | 64.02% | 45.42% | 54.58% | 32.27% | 67.73% |
| Empowerment | 33.33% | 66.67% | 30.88% | 69.12% | 39% | 61% | 30.11% | 69.89% |
| Trustworthiness | 40.75% | 59.25% | 30.58% | 69.42% | 40% | 60% | 31.26% | 68.74% |
| Depth | 28.54% | 71.46% | 26.23% | 73.77% | 37.85% | 62.15% | 30.25% | 69.75% |
| Density | 40.75% | 59.25% | 31.03% | 68.97% | 38.65% | 61.35% | 25.07% | 74.93% |
| Overall Winner | 32.12% | 67.88% | 30.58% | 69.42% | 38.71% | 61.29% | 30.69% | 69.31% |

reasoning ability, as shown in Tab. 4, in the hard-level video understanding task, the multi-modal large language model integrated with AdaVideoRAG demonstrates more significant advantages compared to its original model, and the gap between the two becomes more evident as the task difficulty increases. This result not only confirms the effectiveness of AdaVideoRAG in complex reasoning scenarios but also indirectly validates the rationality and scientific nature of the three-level difficulty division in the HiVU benchmark, providing a reliable basis for quantitatively evaluating the reasoning ability of models.

Meanwhile, we conducted a horizontal comparison with VideoRAG [32] in the HiVU benchmark, as shown in Tab. 4. Consistent with our expectations, AdaVideoRAG is on par with VideoRAG [32] at the Level-1 and Level-2 levels. However, our method exhibits more prominent advantages at the Level 3 which need global and multi-hop reasoning.

Table 4: **Performance on HiVU**. **Left:** Results comparison w/o and w/ AdaVideoRAG. **Right:** Results comparison w/ VideoRAG [32] and AdaVideoRAG.

| Metric | Level-2 | | Level-3 | | Overall | | Level-2 | | Level-3 | | Overall | |
|---|---|---|---|---|---|---|---|---|---|---|---|---|
| | VideoLLaMA3 | VideoLLaMA3 w/ AdaVideoRAG | VideoLLaMA3 | VideoLLaMA3 w/ AdaVideoRAG | VideoLLaMA3 | VideoLLaMA3 w/ AdaVideoRAG | VideoRAG [32] | VideoLLaMA3 w/ AdaVideoRAG | VideoRAG [32] | VideoLLaMA3 w/ AdaVideoRAG | VideoRAG [32] | VideoLLaMA3 w/ AdaVideoRAG |
| Comprehensiveness | 42.72% | 57.28% | 26.00% | 74% | 35.98% | 64.02% | 47.21% | 52.79% | 41.23% | 58.77% | 45.33% | 54.67% |
| Empowerment | 36.81% | 63.19% | 25.11% | 74.89% | 30.88% | 69.12% | 45.18% | 54.82% | 43.81% | 57.05% | 43.81% | 56.19% |
| Trustworthiness | 36.81% | 63.19% | 26.45% | 73.55% | 30.58% | 69.42% | 48.1% | 51.9% | 43.87% | 56.13% | 46.4% | 53.6% |
| Depth | 34.09% | 65.91% | 22.87% | 77.13% | 26.23% | 73.77% | 43.98% | 56.02% | 40.53% | 59.47% | 40.88% | 59.12% |
| Density | 38.63% | 61.37% | 25.11% | 74.89% | 31.03% | 68.97% | 46.36% | 53.64% | 42.56% | 57.44% | 44.1% | 55.9% |
| Overall Winner | 37.27% | 62.73% | 22.87% | 77.13% | 30.58% | 69.42% | 46.17% | 53.83% | 42.23% | 57.77% | 44.1% | 55.9% |

**Ablation Study** In the following analysis, we perform three ablation studies to precisely assess the key components of our proposed method. They are as follows: *1) Without Graph:* We cancel the retrieval of entities and relationships in the graph map; *2) Without vision retrieval:* We remove the feature retrieval in vision-to-text grounding; *3) Without naive text retrieval:* We cancel the the retrieval from the caption, OCR, and ASR databases, as shown in Tab. 5. It can be seen that the design of each module is effective and can improve the understanding ability of the model.

Table 5: Ablation on graph-based knowledge retrieval, vision-based embedding retrieval and auxiliary text retrieval components.

| Metric | w/o Graph | All | w/o Vision | All | w/o Text | All |
|---|---|---|---|---|---|---|
| Comprehensiveness | 38.92% | 61.08% | 50.13% | 49.87% | 33.17% | 66.83% |
| Empowerment | 47.79% | 52.21% | 48.42% | 51.58% | 40.53% | 59.47% |
| Trustworthiness | 47.79% | 52.21% | 46.31% | 53.69% | 39.79% | 60.21% |
| Depth | 46.31% | 53.69% | 49.47% | 50.53% | 30.33% | 69.67% |
| Density | 51.73% | 48.27% | 46.84% | 53.16% | 35.36% | 64.64% |
| Overall Winner | 45.82% | 54.18% | 48.23% | 51.77% | 31.25% | 68.75% |

**Impact of Query Classification.** We conducted two experiments to examine how query classification affects retrieval and reasoning performance. First, we compared different classifier models in terms of classification precision and downstream results. Second, we tested the effect of removing the classifier by routing all queries through the same difficulty level.

**(1) Classifier Comparison.** We evaluated several models on HiVU (with ground-truth Level labels) and the VideoMME benchmark. As shown in Table 6, Qwen2.5-7B achieved the best balance between classification accuracy and overall performance.

**(2) Effect of Removing the Classifier.** We further examined the role of query classification by disabling the classifier and forcing all queries to follow a single retrieval path. When all queries were treated as Level-1, Level-2, or Level-3, the overall scores on the VideoMME benchmark dropped to 64.2, 67.5, and 67.1, respectively. In contrast, our adaptive approach (**AdaVideoRAG**) achieved the

Table 6: Performance of different classifier models.

| Classifier | Precision (HiVU) | Overall Score (VideoMME) |
|------------|------------------|--------------------------|
| Qwen2.5-1.5B | 0.41 | 65.3 |
| Qwen2.5-7B | 0.81 | 68.5 |
| VideoLLaMA3-7B | 0.48 | 67.5 |

highest score of **68.5**, demonstrating that dynamic query routing effectively balances reasoning depth and efficiency.

## 3.3 Efficiency Analysis

To evaluate the efficiency of different retrieval paths, we conducted time-cost experiments on 100 randomly selected videos from the MLVU dataset under three settings: no retrieval (Level-1), simple retrieval (Level-2), and hard retrieval (Level-3).

**(1) Database Construction.** Level-1 performs direct inference without database construction. The average construction time for Level-2 and Level-3 is 351 s and 412 s, respectively, with the additional cost in Level-3 attributed to graph construction. The adaptive scheduling strategy of **AdaVideoRAG** effectively reduces such high-cost operations by prioritizing simpler retrieval paths when applicable.

**(2) Single-Process Inference.** On a single H20 GPU, the average response times are 8 s (Level-1), 26 s (Level-2), and 27 s (Level-3). **AdaVideoRAG** achieves a balanced trade-off between accuracy and efficiency, with an average response time of 20 s.

**(3) Parallelization.** To further improve deployment efficiency, we introduced multi-process and multi-GPU parallelism. Using dual processes on a single H20 GPU (96 GB, batch size = 2), database construction for Level-2 and Level-3 achieved $\sim 2\times$ acceleration, reducing the time to 176 s and 210 s. Scaling to 8 GPUs yielded near-linear speedup ($\sim 8\times$), cutting construction time to 22 s and 26 s. For multi-process retrieval on a single GPU, Level-2 and Level-3 achieved $\sim 2\times$ acceleration, with total processing times of 15 s and 16 s, compared to 20 s for AdaVideoRAG.

## 4 Conclusion

Our AdaVideoRAG demonstrates outstanding performance advantages when dealing with difficult video understanding tasks that require multi-hop thinking and deep reasoning. Meanwhile, AdaVideoRAG can efficiently integrate omni-information, fully leverage the value of multi-source data such as images, videos, and texts. It can also flexibly switch between basic question-answering and high-order semantic understanding tasks, effectively balancing efficiency and accuracy, and greatly enhancing the generalization ability and application universality of multi-modal large language models.

**Limitations, broader impact and social impact.** Due to the limited computational resources for local deployment, this study only evaluates models up to 32 billion parameters. Moreover, the research only implemented three levels of routing, while real-world applications may require more detailed classification. From a social impact perspective, this technology could pose new risks such as spreading false information, for example, using generated fake videos to manipulate public opinion. This highlights the urgent need to establish a collaborative governance system that includes technical ethics, legal regulations, and industry standards.

**Acknowledgments and Disclosure of Funding.** This work was supported by the "Leading Goose" Key R&D Program of Zhejiang Province (2025C01069).

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

# Appendix

The appendix presents the following sections to strengthen the main manuscript:

— **Sec.** A shows the Related Work part of the paper.

— **Sec.** B shows the dynamic sampling strategies when building the database. We set different sampling frequencies to evaluate the performance of our AdaVideoRAG on the MLVU dataset.

— **Sec.** C shows the instructions for level classification. We develop hierarchical prompting templates that enable LLMs to classify query complexity into three tiers (L1-L3), triggering adaptive retrieval strategies from direct lookup to sensemaking reasoning.

— **Sec.** D shows the more detailed results, including multiple-choice QA and summary video understanding, that visually demonstrate our AdaVideoRAG can enhance the reasoning ability of MLLMs.

## A  Related Work

### A.1  Multimodal Large Language Models

With the successful application experience of Large Language Models (LLMs) [40, 41, 17, 45], Multimodal Large Language Models (MLLMs) have also made significant breakthroughs in the field of visual language understanding. LLaVA [31], by first conducting instruction fine-tuning training on a dataset carefully selected by GPT-4, has become one of the most popular methods for constructing MLLMs and has been followed by subsequent works [52, 4, 3]. Benefiting from the rapid development in the fields of short video applications and video generation models, video understanding has also become increasingly popular recently [48, 50, 26]. In contrast, due to the complexity of spatio-temporal joint modeling, video understanding poses higher requirements for the fine-grained spatial understanding of the model and multi-hop prompts, so some works targeting the long-video setting have emerged recently. Vamba [35] has constructed a hybrid Mamba-Transformer model that encodes video tokens with linear complexity, and recent VideoChat-R1 [27] has explored the use of GRPO for the reinforcement fine-tuning of video MLLMs. Recently, there are also some works on video understanding based on agents [30, 13, 49]. Closed-source commercial models such as GPT-4o [21] and Gemini [38] have leading video understanding capabilities, while open-source model series such as the typical Qwen-VL [1], InternVL [9], and VideoLLaMA [47] have received widespread attention and academic research. However, when applied to specific fields, MLLMs have lower accuracy in answers or larger hallucinations due to their inability to dynamically update the latest knowledge. To improve the accuracy, additional fine-tuning costs and time costs are required, and it is difficult to deal with multi-hop questions in long-video cross-modal understanding. In order to alleviate the above problems, this paper studies Retrieval-Augmented Generation (RAG) for long-video, and improves the video understanding ability of the model by integrating the collaborative reasoning between an external knowledge base and the generation model.

### A.2  Retrieval-Augmented Understanding

Retrieval-Augmented Generation (RAG) optimizes large language models (LLMs) by integrating external knowledge retrieval with generative model capabilities [25, 34, 12]. It enables low-cost knowledge expansion without retraining the model through dynamic updates to external knowledge bases, effectively mitigating traditional LLMs' issues of hallucinations, outdated knowledge, and data security risks. RAG [28, 15] converts user queries into vectors, retrieves the most relevant information from external databases, and integrates retrieval results as context into the generative model's prompt to deliver more accurate, reliable, and fact-based responses. However, these methods often overlook complex relationships between documents or contexts—such as entity connections, hierarchical structures, or causal relationships—that are critical for contextual understanding. Consequently, graph-based RAG methods [10] have gained traction, exploring structured knowledge representations to enhance retrieval efficiency and precision while excelling in query-focused summarization tasks. Additionally, Adaptive-RAG [23] significantly improves traditional RAG systems' performance and efficiency in complex scenarios by dynamically adjusting retrieval strategies and generative logic.

Recent research has deeply investigated efficiency challenges [19, 11], dataset-specific optimizations [24], and hallucination correction [44].

Recent RAG approaches have also integrated multimodal information to meet growing application demands [20, 51], such as images [7], code [42], tables [6], and audio [8]. However, constrained by video data's complex modal information and requirements for spatio-temporal modeling, RAG has seen limited adoption in the video understanding field—particularly for long-video understanding, which poses significant challenges for long-context information modeling. Luo *et al.* [32] incorporates visually-aligned auxiliary text features from OCR, ASR, and object detection to construct video knowledge bases, while VideoRAG [36] significantly improves the accuracy of long-video contextual information by building graph databases. However, current solutions either struggle to address multi-hop questions effectively or face substantial computational resource and time costs when maintaining hierarchical graph databases. To tackle these issues, we propose a query intent classification strategy to adaptively route queries to different retrieval paths based on their difficulty, achieving a robust balance between performance and efficiency.

## B    Dynamic Sampling Strategies

During the database construction phase, whether it is naive vector retrieval or graph retrieval, both of them are constructed upon the extracted auxiliary texts, including the caption, ASR, and OCR. Therefore, dynamic sampling strategies directly determine the semantic coverage density. Generally speaking, the faster the sampling frequency, the more effective information can be obtained from the video, which is more conducive to subsequent retrieval.

To systematically validate the impact of sampling strategies, we conduct ablation studies comparing frame sampling density (varying from 5 to 30 frames/clip), measuring their effects on MLVU test dataset, as shown below Tab. A1. As the sampling frequency increases, the evaluation metrics will become higher. Since the final results of 30 frames and 5 frames only differ by about 1 point, we adopt a sampling frequency of 5 frames per 30 seconds for AdaVideoRAG, considering both the improvement of accuracy and the efficiency of resource utilization.

Table A1: The accuracy of different sampling frequencies for the MLVU dataset

| Frames | 5 | 10 | 15 | 20 | 25 | 30 |
|--------|------|------|------|------|------|------|
| M-AVG  | 53.2 | 53.7 | 53.5 | 54.1 | 54.1 | 54.5 |

## C    Instructions for Level Classification

In this paper, we use prompts to classify the difficulty level of queries in Fig. A2, including the detailed description of each level: L1 (Direct Factual) queries requiring single-modality pattern matching (e.g., "Identify the object at 0:05"), L2 (Contextual) needing temporal/causal reasoning across clips (e.g., "Why did X occur before Y?"), and L3 (Multi-Hop) demanding cross-modal hypothesis validation with external knowledge (e.g., "How would Z change if scene A were removed?")

## D    More Detailed Results.

To further validate the enhancement of AdaVideoRAG on the video understanding capabilities of MLLMs, we provide more detailed and specific instance demonstrations, including multiple-choice questions and sensemaking questions, as shown as Fig. A1.

We selected two specific multiple-choice cases and compared them with the MLLM baseline model (VideoLLaMA) and VideoRAG [32] respectively. The results show that our AdaVideoRAG pays more attention to details in terms of video understanding capabilities and thus makes more accurate and confident responses to user queries. Meanwhile, we also analyze the sensemaking problems. As shown at the bottom of Fig. A1, our AdaVideoRAG can provide more fine-grained information. All the answers are based on the real video content, which significantly enhances its ability to reduce visual hallucinations and enables it to output more complete and logical answers.

**multiple-choice questions**

**Query: How many people jump into the water first at the beginning of the video?**
       **(A) 7   (B) 3   (C) 8   (D) 2   (E) 5   (F) 4**

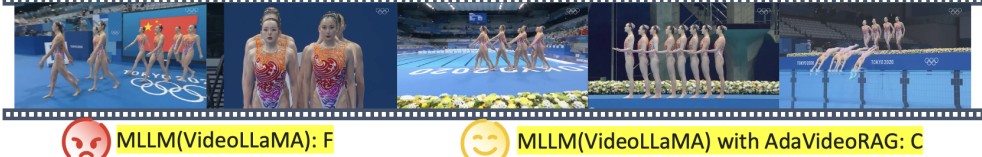

    MLLM(VideoLLaMA): F        MLLM(VideoLLaMA) with AdaVideoRAG: C

**Query: What is the genre of this video?**
        **(A) It is a news report that introduces the history behind Christmas decorations.**
        **(B) It is a documentary on the evolution of Christmas holiday recipes**
        **(C) It is a travel vlog exploring Christmas markets around the world**
        **(D) It is a tutorial on DIY Christmas ornament crafting**

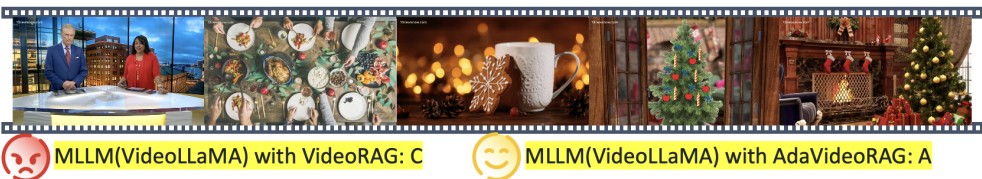

MLLM(VideoLLaMA) with VideoRAG: C      MLLM(VideoLLaMA) with AdaVideoRAG: A

**Sensemaking questions**

  **Query: How does Jean's makeup routine reflect her personal style and preferences?**

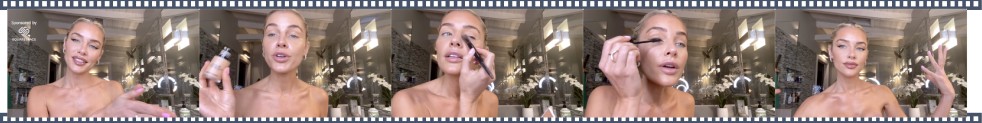

**MLLM(VideoLLaMA) :** Jean's makeup routine reflects her personal style and preferences through her choice of products, application techniques, and overall aesthetic. She opts for a natural yet polished look, using a mix of drugstore and high-end products. Her preference for a minimalist approach is evident in her simple eye makeup and the use of a single lip color.

**MLLM(VideoLLaMA) with AdaVideoRAG :** Jean's makeup routine reflects her personal style and preferences through her emphasis on a 'Clean Girl' look, which focuses on creating a dewy, hydrated appearance without heavy makeup. She prefers using products like the Mac lipstick and the Giorgio Armani luminous silk in the shade 575 for a natural yet polished finish. Her approach to skincare and makeup suggests she values a fresh and radiant look, often achieved through careful application techniques and product selection that enhances her natural features without appearing cakey or overdone.

Figure A1: Qualitative results of VideoLLaMA when applying Video-RAG

**Prompts for Level Classification**

- Goal -

1. Given a query, Classify queries into exclusively one difficulty level (A/B/C).
2. When you are confused about which level to choose, choose a higher level.

#### Classification Criteria ####

**Level 1** (Local Perception or question is very easy that we can put all of video as input)

**Level 2** (Global Search) : It contains simple logical relationships. It can be used for global awareness, which requires global search and positioning of video segments related to questions, and then understanding of the searched video segments.

**Level 3** (Complex Reasoning) : Requires external knowledge/abstract interpretation. It may be necessary to make use of knowledge graphs of multi-layer or complex relationships, or semantic questions with high generalizations

#### Examples ####

Query1: What color is the dog that appears in the 5s part of the video?
Output: ###Level A###
Query2: What happened before the rain?
Output: ###Level B###
Query7: Identify the position relationships of the characters in the meeting room scene?
Output: ###Level C###

Figure A2: Prompts for Level Classification

