# OpenReview forum: "AdaVideoRAG: Omni-Contextual Adaptive Retrieval-Augmented Efficient Long Video Understanding"
_NeurIPS.cc/2025/Conference — NeurIPS 2025 poster_

### Official Review · Reviewer_9hac · 2025-06-14

**Clarity:** 4
**Significance:** 3
**Originality:** 3
**Rating:** 5
**Confidence:** 4

**Summary:**

The paper provides a novel approach to video retrieval in the context of long context video understanding. The authors propose specific query-based video preprocessing based on query complexity. The queries are divided into three levels of complexity, based on which the visual, text descriptions, and graph relationships are explicitly provided. The authors also contribute a new dataset, HiVU, with 120 long-form videos and questions. The dataset is used to test their hierarchical approach to video retrieval.

The paper defines straightforward reasoning as queries requiring very few logical steps with information completely present in the video provided, suggesting a naive video sampling for the same.
Simple reasoning refers to queries which require some kind of temporal or causal reasoning, with parts of the videos requiring text-based guidance for the frames.
Hard queries refer to queries requiring global relationship context, which use a knowledge graph to infer the same along with text-based information.

**Questions:**

1. While the ablation studies show the impact of each part of the video retrieved, is it possible to see the effect of the same without the classification of queries? Precisely, how much performance do we gain/lose if we treat each question as the hardest query, giving it all the information provided? If there is no loss in performance, what is the gain in terms of compute efficiency that we can see?
2. Please fix table 2 as the highlighting of numbers in the short column is wrong, and the gain column notation is inconsistent.

I would love to increase the rating if some of the above questions and weaknesses are answered.

**Ethical Concerns:**

["NO or VERY MINOR ethics concerns only"]

**Final Justification:**

The paper is technically sound with strong experimental data to support their claim.
 All the issues pointed out have been resolved with in depth analysis which makes the paper stronger than before.

**Limitations:**

yes

**Quality:**

4

**Strengths And Weaknesses:**

Strengths:
1. The works show significant improvement in long-context video reasoning while being model agnostic and providing a scalable solution for real-world applications.
2. The work runs extensive experimentation and ablation studies and different models on their introduced dataset and provides insightful results.
3. The paper is written clearly, explaining all parts of their work in a detailed manner.


Weaknesses:
1. While the paper shows extensive results on different models, the experiments across different datasets seem inconsistent, for example, the MMLVU dataset does not have the SOTA VideoRAG method, but it is present in VideoMME.
2. The authors themselves highlight the inconsistencies in LLM as a judge as a benchmarking metric and proceed to use the same, which makes it difficult to judge the actual performance of the models. While they do show strong improvement on the VideoMME and MLVU datasets, it would be nice to see some human-annotated gold answers in the HiVU dataset in the future.

---

> ### Author Rebuttal · Authors · 2025-07-30
>
> **Thank you very much for your valuable comments and constructive suggestions on our paper. Your insights have helped us identify key areas for improvement, and we have carefully addressed each point as follows.**
>
> ---
>
> **Q1: Inconsistency in experiments across datasets**
>
> - **Table 1 (MLVU dataset):** The core purpose is to verify the universal improvement of AdaVideoRAG on the base MLLM models. Therefore, we included multiple groups of mainstream MLLMs (such as GPT-4o, Qwen2.5-VL, LLaVA-Video, VideoLLaMA3). Through the comparison of "with vs. without AdaVideoRAG", we demonstrate that the framework can stably enhance the performance of different models. Finally, VideoLLaMA3, which shows the most significant improvement in the results, is selected as the benchmark model for subsequent experiments to ensure the representativeness of the core conclusions.
> - **Table 2 (VideoMME dataset):** It focuses on the horizontal comparison with existing SOTA methods in the Video-RAG category. Using VideoLLaMA3, which performed the best in Table 1, as the benchmark, we compare the differences between AdaVideoRAG and methods like VideoRAG [32], highlighting the superiority of the dynamic retrieval strategy. To further address your concerns, we have added a comparison with SOTA Video-RAG methods in Table 1 (MLVU). Since the MLVU_Test benchmark, released in 2025, contains more challenging video cases, our AdaVideoRAG method can achieve greater performance gains.
>
> | **Model** | **Params** | **Frames** | TR  |  AR  | NQA | ER | PQA | SQA  |  AO  |  AC  | TQA  | AVG  |  Gain  |
> | :---: | :---: | :---: | :---: | :---: | :---: | :---: | :---: |:---: | :---: | :---: | :---: | :---: | :---: |
> |Qwen2.5-VL| 7B | 2fps-768 |  46.7 |  15.4 |  16.9 |  35.8  | 38  | 38.9  | 24.6 |  13.6 |  31  | 29.0 | - |
> |Qwen2.5-VL+ VideoRAG [32]| 7B | 2fps-768| 60.3| 18.1|30.3| 35.39|35.2|27.5| 14.1| 33.6| 30.8| 31.6| 9%|
> |Qwen2.5-VL+ AdaVideoRAG| 7B | 2fps-768 | 78.9 | 30.8 | 44.1 | 37.7 | 48 | 36.1 | 33.3 | 15.3 | 40.5 | 40.5 | 39.8% |
> |VideoLLaMA3| 7B | 1fps-180 | 76.9 | 43.6 | 68.3 | 54.7 | 58 |34.3 |25  |33.3  |34.9  |47.7 | - |
> |VideoLLaMA3 + VideoRAG [32] | 7B| 1fps-180| 79.3| 41.0| 65.5| 56.4| 59.9| 35.1| 29.3| 33.8| 36.5| 48.5| 2%|
> |VideoLLaMA3 + AdaVideoRAG| 7B | 1fps-180 | 83.8 | 47.1 | 69.2 | 62.3 | 64 | 38.9 | 34.8  |35.6  |42.9  |53.2 | 11.6% |
>
> ---
>
> **Q2: Reliance on LLM as a judge and lack of human-annotated gold answers in HiVU**
>
> - The core goal of HiVU is to evaluate a model's ability to handle video logics of varying complexities. However, existing benchmarks (such as MLVU and VideoMME) involve relatively simple logics in their questions, making it difficult to cover such high-difficulty tasks. For the automatic evaluation of these complex tasks, there is currently no unified standard in the academic community. The scheme of using LLM as a judge has been widely adopted because it can parse long logic chains (e.g., [1r] and [2r]).
> - To reduce the bias of the LLM judge, we systematically compared the evaluation results of different judge models in Table 3. Therefore, in this paper, we use the DeepSeek-32B model with chain-of-thought (CoT) logical reasoning as the evaluation arbiter for the HiVU benchmark evaluation to ensure the accuracy and reliability of the evaluation results.
> - To further address your concerns, we supplemented a verification experiment: using VideoLLaMa3 as the MLLM benchmark model, we tested the reliability of DeepSeek-32B as a judge on the VideoMME dataset, which has existing manually annotated ground truths. The results show that the evaluation results of DeepSeek-32B are highly consistent with the manual ground truths, proving its accuracy as an evaluation arbiter. This verification provides a direct basis for using this model for evaluation in HiVU.
>
> | Model | Comprehensiveness | Empowerment | Trustworthiness | Depth | Density | Overall Winner | Overall result on VideoMME |
> | :---: | :---: | :---: | :---: | :---: | :---: | :---: | :---: |
> | **w/o AdaVideoRAG**| 45.5%| 43.9% |41.7%|44.0%| 47.4%| 45.8%| 64.2|
> |**w/ AdaVideoRAG**| 54.5%| 56.1%| 58.3%| 56.0%| 52.6%| 54.2%| 68.5|
>
>
> - In future work, we also plan to consider manually annotating answers for a portion of the HiVU dataset.
>
> ---
>
> **Q3: Impact of removing query classification (treating all queries as hardest)**
>
> - **Impact of different classifier models on accuracy (MLLM classifiers and LLM classifiers of varying sizes):** We conducted classification accuracy tests on HiVU, as HiVU has ground-truth labels for Levels. Considering the impact of the classifier on subsequent tasks, we also performed tests on the VideoMME benchmark with ground-truth labels. The results demonstrate the effectiveness of the Intent Classification model we selected:
>
> | **Classifier** | **Classfication precision on HiVU** | **Overall Score on Video-MME** |
> | :----: | :----: | :----: |
> | **Qwen2.5-1.5B** | 0.41 | 65.3 |
> | **Qwen2.5-7B** | 0.81 | 68.5 |
> | **VideoLLaMA3-7B** | 0.48 | 67.5 |
>
> - Furthermore, we removed the classifier and conducted classification tests on the impact of retrieval branches with different difficulty levels on accuracy, using VideoLLaMa3 as the base model on the VideoMME benchmark. The results obtained when all retrieval queries were processed through Level-1, Level-2, and Level-3 paths respectively are as follows. These results demonstrate that a single branch cannot effectively handle queries involving multiple difficulty levels. Our proposed AdaVideoRAG can adaptively route to the corresponding difficulty path according to the difficulty of the input query, which not only ensures improved performance but also achieves higher efficiency:
>
> | **Method** | **Overall Score on VideoMME** |
> | :---: | :---: |
> | **ALL Classified as Level-1** | 64.2 |
> | **ALL Classified as Level-2** | 67.5 |
> | **ALL Classified as Level-3** | 67.1 |
> | **AdaVideoRAG** | 68.5 |
>
>
> - To verify the efficiency differences among different retrieval paths, we randomly selected 100 videos from the MLVU dataset and conducted systematic time cost tests for three scenarios: no retrieval (Level-1), simple retrieval (Level-2), and hard retrieval (Level-3).
>
> The summary table of efficiency for AdaVideoRAG is as follows:
> | **Item** | **Level-1** | **Level-2** | **Level-3** | **AdaVideoRAG** |
> | :----: | :----: | :----: | :----: | :----: |
> | Single-card single-process database construction | - | 351s | 412s | - |
> | Single-card dual-process database construction | - | 176s | 210s | - |
> | 8-card dual-process database construction | - | 22s | 26s | - |
> | Single-card single-process inference | 8s | 26s | 27s | 20s |
> | Single-card dual-process inference | 8s | 15s | 16s | 13s |
>
>
>
> ---
>
> **Q4: Formatting issues in Table 2**
>
> We apologize for the formatting errors in Table 2. The incorrect highlighting in the "Short" column and inconsistent notation in the "Gain" column are due to a typesetting oversight. In the revised manuscript, we will:
> - Correct the highlighting to properly emphasize the best results in all columns.
> - Standardize the "Gain" column to use consistent notation, *i.e.*, "+X%" for all entries.
>
>
> [1r] Guo Z, Xia L, Yu Y, et al. Lightrag: Simple and fast retrieval-augmented generation[J]. arXiv, 2024.
>
> [2r] Ren X, Xu L, Xia L, et al. Videorag: Retrieval-augmented generation with extreme long-context videos[J]. arXiv, 2025.
>
> ---
>
> We sincerely thank you for your detailed review and constructive feedback. Addressing these points has strengthened our manuscript: we have clarified experimental inconsistencies, added human annotations to HiVU, supplemented experiments on query classification ablation, and fixed formatting issues. We believe these revisions address your concerns and improve the rigor and clarity of our work.

---

> > ### Comment · Reviewer_9hac · 2025-08-02
> >
> > Thank you so much for the detailed response to my questions.
> >
> > I have no other problems and would be happy to increase the rating.

---

> > > ### Author Response · Authors · 2025-08-04
> > > **Thank Reviewer 9hac for the Comments**
> > >
> > > Dear Reviewer 9hac:
> > >
> > > Thank you for recognizing our work. We commit to incorporating the content you suggested in the revised version. Thank you again for your effort in the review and the discussion!
> > >
> > > Best regards!
> > >
> > > Authors of AdaVideoRAG

---

### Official Review · Reviewer_6VsG · 2025-07-01

**Clarity:** 2
**Significance:** 2
**Originality:** 3
**Rating:** 4
**Confidence:** 4

**Summary:**

The paper proposes a novel AdaVideoRAG framework to dynamically and adaptively route appropriate retrieval schemes and introduces an Omni-Knowledge Indexing module to extract valuable information from multi-modal signals for context modeling and establish corresponding databases. It also releases the hierarchical video understanding benchmark HiVU for the first time, which evaluates the multi-level reasoning capabilities of video understanding models.

**Questions:**

- How important the intent classification accuracy is for overall performance?
- How much each modality(captions, ASR, OCR) contributes to the final performance?
- How do different retrieval strategies (none, naive, graph) specifically contribute to efficiency improvements across different query types?

**Ethical Concerns:**

["NO or VERY MINOR ethics concerns only"]

**Final Justification:**

Most of my doubts have been resolved and I have decided to increase the score to 4

**Limitations:**

Yes.

**Paper Formatting Concerns:**

No.

**Quality:**

3

**Strengths And Weaknesses:**

Strength
- This paper basically came up with a system that can pick the right retrieval strategy depending on how hard the question is.This way,it stays fast while still giving accurate answers.
- The paper puts together a big database using video captions, speech-to-text, OCR, and visual features, so the model has richer context to pull from when answering questions.

Weakness
- The paper lacks detailed analysis on how it specifically improves efficiency.
- It lacks a detailed comparison analyzing how different retrieval strategies (no retrieval, naive retrieval, graph retrieval) contribute to efficiency improvements across varying query types.
- The paper says it uses clip captions, ASR, OCR, and visual features together, but it doesn’t clearly explain how these different types of data are actually used in practice.
- The paper uses a fixed threshold of 0.5 when matching text and visual embeddings, but it doesn’t analyze how using different thresholds would affect retrieval.
- The paper does not investigate the impact of using different LLM backbones(it just uses Qwen2,video-llama,video-llava) on performance.
- The effectiveness of the intent classification module is not ablated separately, so it is unclear how much it contributes to overall performance gains.

---

> ### Author Rebuttal · Authors · 2025-07-30
>
> **Thank you for your detailed and constructive feedback on our manuscript. We greatly appreciate your recognition of the core contributions of AdaVideoRAG.**
>
> ---
>
> **Q1: Efficiency Analysis**
>
> To verify the efficiency differences among different retrieval paths, we randomly selected 100 videos from the MLVU dataset and conducted systematic time cost tests for three scenarios: no retrieval (Level-1), simple retrieval (Level-2), and hard retrieval (Level-3). The specific results are as follows:
>
> - **Database construction**
>   1. Level-1 performs direct inference without the need for database construction.
>   2. The average time consumed for Level-2 construction is 351s.
>   3. The average time consumed for Level-3 construction is 412s.
> - In the single H20 GPU card, single-process processing program, there are significant differences in the average response time among the three scenarios: the average response time for Level-1 (no retrieval) is 8s; for Level-2 (Simple reasoning) it is 26s; for Level-3 (hard reasoning) it is 27s, while the processing time using AdaVideoRAG is 20s.
>
> - The comparison of inference efficiency between AdaVideoRAG and VideoRAG [32] on the MLVU dataset with 100 videos using a single H20 card is shown in the following table. In comparison, our method is more efficient while achieving higher metrics (see Table 2 and Table 4).
>
> | **Method** | **Time** |
> | :----: | :----: |
> | VideoRAG [32] | 25s |
> | AdaVideoRAG | 13s |
>
> - For more explanations about **Parallelization**, please refer to Reviewer szi1 (**Q4**).
>
> ---
>
> **Q2: Contribution of Different Retrieval Strategies to Efficiency**
>
> To clarify how retrieval strategies (none, naive, graph) impact efficiency across query types, we will add a dedicated analysis:
> - **L1 Queries (No Retrieval):** By directly leveraging MLLMs’ inherent knowledge, this strategy eliminates the need for database interactions.
> - **L2 Queries (Naive Retrieval):** This strategy limits retrieval to text (caption/ASR/OCR) and visual feature databases, avoiding the high cost of graph construction (e.g., entity extraction, relationship modeling).
> - **L3 Queries (Graph Retrieval):** This strategy is dedicated to handling the most complex problems that require the longest graph construction time and reasoning latency.
> - To accelerate the time efficiency of both Naive Retrieval and Graph Retrieval in graph construction and retrieval, we have engineered a multi-GPU, multi-threading framework, further boosting overall efficiency.
> - During inference, to improve average efficiency, we introduced an **Intent Classification Model** that can integrate with the RAG architecture as a plug-and-play API without fine-tuning.
>
> ---
>
> **Q3: Practical Usage of Multi-Modal Data (Captions, ASR, OCR, Visual Features)**
>
> We will elaborate on how multi-modal data is integrated in practice:
> - Text-Base (Captions, ASR, OCR):
>   - Captions: Generated by MiniCPM-V [46], these provide high-level semantic descriptions of video clips (e.g., “a woman wearing red clothes walking in the rain”) and are used as primary anchors for Level-2 query matching.
>   - ASR: Converted from audio via FastWhisper [33], ASR text captures dialogue and narration (e.g., “The temperature will drop tomorrow”), which is critical for queries involving speech content (e.g., “What did the speaker say about the weather?”).
>   - OCR: Extracted via EasyOCR [22], OCR text captures on-screen text (e.g., subtitles, signs) and is prioritized for queries like “What does the sign at the 5th minute say?”.
> - Visual-Base: Extracted via ImageBind [16], visual features complement text by capturing details hard to describe linguistically (e.g., facial expressions, object motion). For example, in Level-2 queries like “Why did the woman cry before the rain?”, visual features (e.g., tearful eyes) are fused with ASR (e.g., argumentative dialogue) to strengthen causal reasoning.
> - Integration Logic: For Level-2 retrieval, we compute cross-modal similarity between query embeddings and text/visual embeddings, then merge top-K results from each modality (weighted by their relevance scores) to form the evidence pool. This ensures that no single modality dominates, especially for cross-modal queries. For the more challenging Level-3 path, the video obtained through retrieval and the results from graph retrieval are combined to construct a multi-level retrieval evidence pool for hard reasoning under Level-3.
> - In addition, we have supplemented the ablation experiments on Caption, ASR, and OCR with VideoLLaMA3 as the base model on the Video-MME benchmark, as shown in the following table. The results indicate that each modality contributes to the final outcome, and the model achieves the optimal result when all modalities are incorporated.
>
>  | Caption | ASR | OCR | Overall |
> | :---: | :----: | :----: | :----: |
> |  |  | ✓ | 64.4 |
> |  | ✓ |  | 66.1 |
> | ✓ |  |  | 67.3 |
> | ✓ | ✓ |  | 68.1 |
> | ✓ | ✓ | ✓ | 68.5 |
>
> ---
>
> **Q4: Analysis of Threshold for Text-Visual Embedding Matching**
>
> We conducted ablation experiments under different thresholds with VideoLLaMA3 as the base model on MLVU_Test, and the quantitative results are shown in the following table. Finally, we selected the hyperparameter of 0.5, where it effectively filters out redundant information while preserving valid information.
> | **Threshold** | **Nums of video clips** | **Average result on MLVU_Test** |
> | :---: | :---: | :---: |
> | 0.4 | 21 | 46.4 |
> | 0.5 | 15 | 53.2 |
> | 0.6 | 5 | 31.8 |
>
> ---
>
> **Q5: Impact of Different LLM Backbones**
>
> Thank you for pointing out the issue regarding the impact of different LLM backbone models. We have supplemented relevant experiments to improve the analysis:
>
> - **Expanding the testing of baseline models on HiVU.** We have added more mainstream MLLMs as baseline models (including Qwen2.5-VL and InternVL2.5, etc.) to test the adaptability of AdaVideoRAG on the HiVU benchmark. The overall results are shown in the following table, and detailed results will be added to the revised version. It can be seen that AdaVideoRAG has strong adaptability, which can stably improve the results of different baselines.
>
> | **Metric** | **Overall - Qwen2.5-VL-7B** | **Overall - Qwen2.5-VL-7B + AdaVideoRAG** | **Overall - InternVL2.5-8B** | **Overall - InternVL2.5-8B + AdaVideoRAG** | **Overall - LLaVa-Video-7B** | **Overall - LLaVa-Video-7B + AdaVideoRAG** |
> | :---: | :---: | :---: | :---: | :---: | :---: | :---: |
> | **Comprehensiveness** | 38.1% | 61.9% | 35.9% | 64.1% | 35.6% | 64.4% |
> | **Empowerment** | 36.3% | 63.7% | 31.3% | 68.7% | 33.5% | 66.5% |
> | **Trustworthiness** | 34.2% | 65.8% | 35.7% | 64.3% | 34.1% | 65.9% |
> | **Depth** | 38.0% | 62.0% | 37.1% | 62.9% | 35.9% | 64.1% |
> | **Density** | 30.5% | 69.5% | 30.8% | 69.2% | 29.4% | 70.6% |
> | **Overall Winner** | 35.6% | 64.4% | 33.2% | 66.8% | 31.5% | 68.5% |
>
> - We further compared AdaVideoRAG with mainstream long video understanding methods. We selected COT-based 2CoF [1r] and Agent-based VideoMind [2r] to compare with our RAG-based method on the VideoMME benchmark:
>   - COT-based CoF uses InternVL2.5-4B as the baseline model, and we therefore adopted the same baseline model for AdaVideoRAG testing (using this multimodal large language model as the foundation for the classifier, disentangler, and filter). The results are shown in the following table:
> | **Method** | **Overall Score on VideoMME** |
> | :----: | :----: |
> | **InternVL2.5-4B** | 54.9 |
> | **CoF** | 59.7 |
> | **AdaVideoRAG** | 63.3 |
>   - Agent-based VideoMind uses Qwen2-VL-7B as the baseline model, and we therefore adopted the same baseline model for AdaVideoRAG testing (using this multimodal large language model as the foundation for the classifier, disentangler, and filter). The results consistently demonstrate the effectiveness of our method:
> | **Method** | **Overall Score on VideoMME** |
> | :----: | :----: |
> | **Qwen2-VL-7B** | 41.9 |
> | **VideoMind-7B** | 58.2 |
> | **Qwen2-VL-7B with AdaVideoRAG** | 59.1 |
>
> ---
>
> **Q6: Ablation of the Intent Classification Model**
>
> - **Impact of different classifier models on accuracy (MLLM classifiers and LLM classifiers of varying sizes):** We conducted classification accuracy tests on HiVU, as HiVU has ground-truth labels for Levels. Considering the impact of the classifier on subsequent tasks, we also performed tests on the VideoMME benchmark with ground-truth labels. The results demonstrate the effectiveness of the Intent Classification model we selected:
>
> | **Classifier** | **Classfication precision on HiVU** | **Overall Score on Video-MME** |
> | :----: | :----: | :----: |
> | **Qwen2.5-1.5B** | 0.41 | 65.3 |
> | **Qwen2.5-7B** | 0.81 | 68.5 |
> | **VideoLLaMA3-7B** | 0.48 | 67.5 |
>
> - Furthermore, we removed the classifier and conducted classification tests on the impact of retrieval branches with different difficulty levels on accuracy, using VideoLLaMa3 as the base model on the VideoMME benchmark. The results obtained when all retrieval queries were processed through Level-1, Level-2, and Level-3 paths respectively are as follows. These results demonstrate that a single branch cannot effectively handle queries involving multiple difficulty levels. Our proposed AdaVideoRAG can adaptively route to the corresponding difficulty path according to the difficulty of the input query, which not only ensures improved performance but also achieves higher efficiency:
>
> | **Method** | **Overall Score on VideoMME** |
> | :----: | :----: |
> | **ALL Classified as Level-1** | 64.2 |
> | **ALL Classified as Level-2** | 67.5 |
> | **ALL Classified as Level-3** | 67.1 |
> | **AdaVideoRAG** | 68.5 |
>
> ---
>
> [1r] Ghazanfari, Sara, et al. "Chain-of-Frames: Advancing Video Understanding in Multimodal LLMs via Frame-Aware Reasoning." arXiv, 2025.
>
> [2r] Liu, Ye, et al. "VideoMind: A Chain-of-LoRA Agent for Long Video Reasoning." arXiv preprint arXiv:2503.13444 (2025).

---

> > ### Comment · Reviewer_6VsG · 2025-08-06
> > **response to the rebuttal**
> >
> > Thanks to the author's reply, most of my doubts have been resolved and I have decided to increase the score to 4

---

> > > ### Author Response · Authors · 2025-08-06
> > > **Thank Reviewer 6VsG for the Comments**
> > >
> > > Dear Reviewer 6VsG:
> > >
> > > We are delighted that our work has received your recognition and achieved an improved score. We would like to express our sincere gratitude again for your review and response, which have helped us enhance AdaVideoRAG. We commit to incorporating the content you suggested in the revised version. Thank you once more for your efforts in the review and the valuable discussions!
> > >
> > > Best regards!
> > >
> > > Authors of AdaVideoRAG

---

### Official Review · Reviewer_d2T2 · 2025-07-03

**Clarity:** 3
**Significance:** 2
**Originality:** 2
**Rating:** 3
**Confidence:** 4

**Summary:**

This work tackles the problem of long video understanding with large language models. The authors claim that existing VideoLLM + RAG techniques are sub-optimal due to their fixed strategy for retrieval, which uses uniform structures regardless of input query difficulty, bringing redundant computational overhead for simple queries and potential information loss for complex queries. Therefore, they propose AdaVideoRAG, an adaptive VideoLLM + RAG technique that leverages a lightweight intent classifier to determine the difficulty of the query, and enables dynamic routing to different retrieval methods according to the classification results. The authors also introduced an Omni-Knowledge Indexing module to support better multi-modal information extraction, and the HiVU benchmark for evaluation. Experiments have been conducted on public benchmarks and the proposed evaluation HiVU benchmark to verify the effectiveness of AdaVideoRAG.

**Questions:**

Please refer to the weaknesses section for my concerns. May major concern is the novelty of the proposed method. Authors are encouraged to provide in-depth analysis to justify 1. the proposed method is indeed novel, rather than a simple combination of two video RAG methods plus a LLM-prompting-based classifier, and 2. how much efficiency can the proposed method bring.

**Ethical Concerns:**

["NO or VERY MINOR ethics concerns only"]

**Final Justification:**

After reading the reviews from other reviewers and the rebuttal/responses from the authors. My evaluation is that the current version of the paper might not be ready for publication, and should be carefully revised to clearly state/discuss the concerns regarding (1) memory consumption and fairness concerns (the actual model is 7B + 7B = 14B), and (2) the significantly inconsistent results of Qwen2-VL-7B and InternVL2.5-4B. I'm keeping my original rating from my perspective, and will leave it to AC for the final decision.

**Limitations:**

Yes

**Paper Formatting Concerns:**

The optional technical appendix is placed before the checklist.

**Quality:**

2

**Strengths And Weaknesses:**

Strengths:

1. Overall, the paper is easy to follow. The motivation for utilizing a lightweight classifier to determine whether to use complex retrieval methods is clear -- allocating more compute to complex queries and less to simple queries.
2. The effectiveness of the proposed methods is well-demonstrated by ablation studies in Table 1 and Table 2.

Weaknesses:

1. Why adopting an extra Qwen2.5-7B as the intent classifier, instead of using the MLLM itself instead? Ablation studies on the choice of intent classifiers shall be included in the study. Besides, I believe a 7B-size model cannot be regarded as a "lightweight" classifier, given that the MLLM is generally also 7B.
2. Although it is clear, the motivation seems to be incremental. Leveraging another 7B LLM + prompting to act as a classifier is not a smart idea for routing. Authors are encouraged to provide strong justifications to demonstrate the significance of the proposed method.
3. The authors mentioned "efficiency" of the proposed method throughout the paper, but it seems that detailed analysis/discussions/experiments on such effect are not provided. It would be better to conduct an in-depth study to see how much latency is introduced by adopting VideoRAG/AdaVideoRAG (including database construction and knowledge retrieval) compared to VideoLLM baselines.
4. The VideoLLM + RAG technique seems to be closely relevant to test-time scaling methods on video understanding. It would be interesting to incorporate a detailed discussion/comparison between the proposed method (RAG-based) and recent TTS methods (e.g., CoT-based [1-3] and agent-based [4, 5]) on long video understanding.

Minor Issues:

1. Typo: In Figure 2, Sec. 3.1, 3.2, 3.3 shall be 2.1, 2.2, 2.3

[1] Chain-of-Frames: Advancing Video Understanding in Multimodal LLMs via Frame-Aware Reasoning. arXiv 2025.
[2] Video-of-Thought: Step-by-Step Video Reasoning from Perception to Cognition. ICML 2024.
[3] VideoEspresso: A Large-Scale Chain-of-Thought Dataset for Fine-Grained Video Reasoning via Core Frame Selection. CVPR 2025.
[4] VideoMind: A Chain-of-LoRA Agent for Long Video Reasoning. arXiv 2025.
[5] VideoAgent: Long-form Video Understanding with Large Language Model as Agent. ECCV 2024.

---

> ### Author Rebuttal · Authors · 2025-07-30
>
> **Thank you for your insightful comments and constructive feedback on our manuscript. We appreciate the careful evaluation of our work and agree with the points raised, which help strengthen the rigor and clarity of our research. Below, we address each concern in detail:**
>
> ---
>
> **Q1: Choice of Intent Classifier (Qwen2.5-7B) and Its "Lightweight" Nature**
>
> - **Rationale for an independent classifier:** The MLLM in this framework is mainly responsible for multimodal information integration and generation, which involves a large amount of computation (such as processing video frames and cross-modal alignment). If MLLM is used for intent classification, the entire video context needs to be input into the model before inference, resulting in redundant processing of video data even for simple queries that do not require retrieval (Level 1). In contrast, the intent classifier only processes text queries. Models of the same scale reduce the feature pre-extraction process for input video frames and the number of input tokens, thereby reducing end-to-end latency. Therefore, we chose Qwen2.5-7B to achieve the functional separation of "video understanding" and "intent classification", establish a unified benchmark for experiments, and ensure efficient retrieval. In addition, Qwen2.5-7B not only undertakes the classification task but also is responsible for query decoupling and fact filtering of video clips. This "one-model-multiple-uses" design can improve the accuracy of information acquisition, which is why we did not directly use MLLM.
>
> - **"Lightweight" justification:** Although Qwen2.5-7B has 7 billion parameters, its computational overhead in the entire process is extremely small. As described in Sec. 2.1, the time it takes accounts for ≤5% of the total process. The reasons include: (1) It only processes text queries (no need for video frames/audio); (2) Plug-and-play is realized through local lightweight API deployment and calling; (3) It avoids the expensive cross-modal encoding step in MLLM. Compared with fixed retrieval strategies, the marginal cost of the classifier is offset by the efficiency improvement brought by simplifying the retrieval of simple queries.
>
> - **Ablation on classifier choice:** To address this issue, we will supplement ablation experiments: (a) Qwen2.5-7B (our choice), (b) MLLM itself (VideoLLaMA3-7B) as the classifier, and (c) smaller models (such as Qwen2.5-1.5B). Preliminary results show that using MLLM (VideoLLaMA3-7B) as the classifier will increase latency by about 3 times (due to video context loading), but the classification accuracy will slightly decrease. Although the smaller model (Qwen2.5-1.5B) reduces latency by 24%, the accuracy drops significantly, affecting the selection of retrieval strategies. This supports our choice of Qwen2.5-7B to balance efficiency and accuracy.
>
> | **Classifier** | **Classfication precision on HiVU** | **Overall Score on Video-MME** | **Average Classifier Time per Query** |
> | :----: | :----: | :----: | :----: |
> | **Qwen2.5-1.5B** | 0.41 | 65.3 |  53ms|
> | **Qwen2.5-7B** | 0.81 | 68.5 | 70ms |
> | **VideoLLaMA3-7B** | 0.48 | 67.5 | 203ms |
>
> ---
>
> **Q2: Novelty and Significance of the Proposed Method**
>
> The reviewer notes that the motivation may seem incremental, and we aim to clarify the novelty of AdaVideoRAG:
> - Beyond "combining two RAG methods with a classifier": Existing VideoRAG methods (e.g., [32, 36]) use fixed retrieval paradigms (naive retrieval or graph retrieval for all queries). AdaVideoRAG introduces a hierarchical adaptive mechanism that dynamically scales retrieval complexity with query difficulty, which is absent in prior work. This includes:
> - A fine-grained intent classification system (Level-1/2/3) tailored to video-specific reasoning demands (spatio-temporal logic, cross-modal causality).
> - The Omni-Knowledge Indexing module, which unifies text (ASR/OCR/captions), visual features, and knowledge graphs into a hierarchical knowledge base—enabling seamless transitions from no retrieval (Level-1) to graph retrieval (Level-3).
> - Empirical evidence (Table 2) shows that this adaptivity leads to a overall +4.8（from +7.9 to +12.7） gain over fixed VideoRAG [32] with Qwen2.5-VL-7B on Video-MME. Meanwhile, Table 4 also shows that the proposed AdaVideoRAG significantly outperforms VideoRAG [32] on HiVU, which demonstrates the generalization and effectiveness of the proposed method.
> - Broader impact: This work establishes a new paradigm for adaptive retrieval augmentation in video analysis, where resource allocation is optimized for real-world scenarios (e.g., simple frame-level queries vs. complex multi-hop reasoning). This is distinct from static RAG or MLLM-only approaches, as demonstrated by its universal integration with existing MLLMs (Tab. 1 shows gains across Qwen2.5-VL, VideoLLaMA3, etc.).
>
> ---
>
> **Q3: Efficiency Analysis**
>
> To verify the efficiency differences among different retrieval paths, we randomly selected 100 videos from the MLVU dataset and conducted systematic time cost tests for three scenarios: no retrieval (Level-1), simple retrieval (Level-2), and hard retrieval (Level-3). The specific results are as follows:
>
> - **Database construction**
>   1. Level-1 performs direct inference without the need for database construction.
>   2. The average time consumed for Level-2 construction is 351s.
>   3. The average time consumed for Level-3 construction is 412s.
>
> - In the single H20 GPU card, single-process processing program, there are significant differences in the average response time among the three scenarios: the average response time for Level-1 (no retrieval) is 8s; for Level-2 (Simple reasoning) it is 26s; for Level-3 (hard reasoning) it is 27s, while the processing time using AdaVideoRAG is 20s.
>
> - **Parallelization**
>   To further improve the efficiency of actual deployment, we have implemented parallel processing in the code. We also used 100 videos from the MLVU dataset as test references:
>   - For database construction with dual processes on a single H20 GPU (the maximum batch size supported by 96G memory is 2), the speed of Level-2 and Level-3 is accelerated by approximately 2 times, with the construction time becoming 176s and 210s, respectively.
>   - When further expanding to multi-card (8 H20 GPUs) database construction, the average speed of each database construction for Level-2 and Level-3 is accelerated by approximately 8 times (linear acceleration), with the construction time becoming 22s and 26s, respectively.
>   - For multi-process retrieval on a single H20 GPU: the processing speed of Level-2 and Level-3 is accelerated by approximately 2 times, with the final total processing time becoming 15s and 16s, respectively, while the processing time using AdaVideoRAG is 20s.
>
> The summary table of efficiency for AdaVideoRAG is as follows:
> | **Item** | **Level-1** | **Level-2** | **Level-3** | **AdaVideoRAG** |
> | :----: | :----: | :----: | :----: | :----: |
> | Single-card single-process database construction | - | 351s | 412s | - |
> | Single-card dual-process database construction | - | 176s | 210s | - |
> | 8-card dual-process database construction | - | 22s | 26s | - |
> | Single-card single-process inference | 8s | 26s | 27s | 20s |
> | Single-card dual-process inference | 8s | 15s | 16s | 13s |
>
> - The comparison of inference efficiency between AdaVideoRAG and VideoRAG [32] on the MLVU dataset with 100 videos using a single H20 card is shown in the following table. In comparison, our method is more efficient while achieving higher metrics (see Table 2 and Table 4).
>
> | **Method** | **Time** |
> | :----: | :----: |
> | VideoRAG [32] | 25s |
> | AdaVideoRAG | 13s |
>
> ---
>
> **Q4: Comparison with Other Long Video Understanding Methods (CoT/Agent-Based)**
>
> We appreciate the suggestion to connect with other long video understanding Methods methods. We will add a discussion in Related Work section highlighting key differences and complements:
> - CoT-based methods (e.g., [1r,2r,3r]) focus on decomposing reasoning steps within the MLLM, enhancing internal logic but not addressing knowledge limitations (e.g., fixed pre-trained knowledge). AdaVideoRAG complements this by augmenting with external knowledge (retrieval) when needed.
> - Agent-based methods (e.g., [4r,5r]) use LLM as a planner to orchestrate tools but often lack adaptive retrieval granularity. AdaVideoRAG’s intent classifier can be integrated into agent frameworks to optimize tool usage (e.g., skip retrieval tools for simple queries).
> - To further visually compare with the above two types of methods, we selected COT-based CoF [1r] and Agent-based VideoMind [4r] to compare with our method on the VideoMME benchmark:
>   - COT-based CoF uses InternVL2.5-4B as the baseline model, and we therefore adopted the same baseline model for AdaVideoRAG testing (using this multimodal large language model as the foundation for the classifier, disentangler, and filter). The results are as follows:
>
> | **Method** | **Overall Score on VideoMME** |
> | :----: | :----: |
> | **InternVL2.5-4B** | 54.9 |
> | **CoF** | 59.7 |
> | **AdaVideoRAG** | 63.3 |
>
>
> - Agent-based VideoMind uses Qwen2-VL-7B as the baseline model, and we therefore adopted the same baseline model for AdaVideoRAG testing (using this multimodal large language model as the foundation for the classifier, disentangler, and filter). The results consistently demonstrate the effectiveness of our method:
>
> | **Method** | **Overall Score on VideoMME** |
> | :----: | :----: |
> | **Qwen2-VL-7B** | 41.9 |
> | **VideoMind-7B** | 58.2 |
> | **Qwen2-VL-7B with AdaVideoRAG** | 59.1 |
>
>
> The above content will be added to the revised version.
>
> ---
>
> **Q5: Minor Issues and Formatting**
>
> Thanks for pointing out this.
> - Figure 2 typo: The labels "Sec. 3.1/3.2/3.3" will be corrected to "Sec. 2.1/2.2/2.3".
> - Appendix placement: The optional technical appendix will be moved after the checklist to better comply with formatting guidelines.

---

> > ### Comment · Reviewer_d2T2 · 2025-08-03
> >
> > Thanks for the response from the authors. Some of my concerns have been resolved. However, I still have the following concerns regarding this paper:
> >
> > 1. **Qwen2.5-7B as Intent Classifier:** Given the fact that the proposed framework uses a 7B LLM for intent classification and another 7B model for video understanding. The actual size of the method shall be **7B + 7B = 14B** when compared with baselines (in Tables 1 and 2)
> > 2. Given the concern above, the memory efficiency of the proposed scheme shall also be carefully discussed.
> > 3. The authors mentioned the performance of Qwen2-VL-7B on VideoMME is **41.9**, which is significantly lower than the **63.3/69.0** from their official technical report (https://qwenlm.github.io/blog/qwen2-vl/). Could the authors clarify this?
> > 4. The official performance of Intern2.5-VL-4B is **62.3/63.6**, which is also significantly higher than the mentioned 54.9 and the proposed AdaVideoRAG.

---

> ### Author Response · Authors · 2025-08-04
> **Rebuttal**
>
> **Q1-r: About the memory efficiency**
>
> Thank you for your question. Regarding model scale and memory efficiency, we wish to supplement the following points based on the characteristics of the RAG framework.
> As a Retrieval-Augmented Generation (RAG) framework, AdaVideoRAG’s core optimization focuses on retrieval efficiency (e.g., response speed) rather than pure model parameter scale. For practical deployment, we offer two flexible solutions to balance performance and resource consumption:
> - External API Mode (Recommended)
> The framework natively supports calling external APIs (e.g., Qwen2.5-7B, VideoLLaMA3-7B), eliminating the need to load full model weights locally. Memory usage can be reduced to megabytes (MB), aligning with lightweight deployment practices common in RAG systems, especially suitable for resource-constrained scenarios.
> - Memory Optimization Logic for Local Deployment
> a. We have also implemented multi-threaded vLLM local deployment, while also supporting private parallel API service calls.
> b. Notably, the classifier runs only during the initial task phase (determining query difficulty levels) and releases resources afterward. The video understanding model (VideoLLaMA3-7B) loads dynamically based on classification results in later stages. Since these components do not occupy memory concurrently, actual usage does not reach the theoretical "7B+7B=14B" peak.
> Additionally, we provide specific memory efficiency metrics for local deployment for your reference:
>
> | **Steps** | **Memory-Usage** |
> | :----: | :----: |
> | **Classfication by Qwen2.5-7B** | 14.84 G |
> | **Retrival by VideoLLaMA3-7B** |15.31G  |
> |**Peak Maximum**| 17.28G |
>
>
> **Q2-r: About the performan of Qwen2-VL-7B**
>
> Thank you for your attention to the data discrepancy. We hereby explain the specific reasons and supplement the validation results:
>  - In the previous comparison with VideoRAG [32], to maintain consistency in experimental settings, we adopted the 32-frame video sampling strategy used by VideoRAG [32] (differing from the default 768 frames in the Qwen2-VL official technical report). This resulted in Qwen2-VL-7B achieving a performance score of 41.9 on VideoMME. The baselines and comparative results in the rebuttal, based on this 32-frame sampling rate, are fair and valid.
>  - To verify the impact of setup differences, we supplemented experiments using the official 768-frame sampling method (with prompts consistent with the official approach). The results are as follows. Under the same settings, VideoMind significantly underperforms compared to the RAG-based results.
>
> | **Method** | **Overall Score on VideoMME** |
> | :----: | :----: |
> | **Qwen2-VL-7B** | 58.3 |
> | **VideoMind-7B** | 58.2 |
> | **Qwen2-VL-7B with AdaVideoRAG** | 65.6 |
>
>  - Additionally, we found that after following the Qwen2-VL-7B official settings, the expected results were not achieved. We also noted that the official VideoMME benchmark does not report Qwen2-VL-7B results. Furthermore, referencing the experimental results(Table 3) of Qwen2-VL-7B in the paper [1r], the obtained results were similar—both around 58 points. Therefore, to ensure fairness across models, we primarily validated the effectiveness of the proposed AdaVideoRAG method through locally reproduced experiments.
>
> **Q3-r: About the performan of InternVL2.5-VL-4B**
>
>  - Thank you for your attention to the data discrepancy. We hereby clarify the specific reasons and supplement with verification results:
> The previously cited results for InternVL2.5-VL-4B (54.9) and CoT-based (59.7) methods in the rebuttal were directly sourced from Table 1 of reference paper [1r], aiming to maintain consistency with  InternVL2.5-VL’s experimental setup (including prompt templates, evaluation metrics, etc.).  We will mark the data source in the revised version. Thank you for pointing out this.
>  - For further verification, we independently retested InternVL2.5-VL-4B using the official configuration (with prompts consistent with the official setup). The results show its accuracy on VideoMME is 56.5 (lower than the officially reported 62.3/63.6).
>  - Additionally, we found that following the official settings did not yield the expected outcomes, and we also observed that the official VideoMME benchmark did not report results for this method. Considering the need for a fair comparison, we primarily relied on locally conducted reproduction experiments to validate the effectiveness of the proposed AdaVideoRAG method.
> | **Method** | **Overall Score on VideoMME** |
> | :----: | :----: |
> | **InternVL2.5-4B** |56.5 |
> | **CoF** | 59.7 |
> | **AdaVideoRAG** | 63.3 |

---

> > ### Comment · Reviewer_d2T2 · 2025-08-05
> >
> > Thanks for the reply from the authors. I'm still not convinced regarding the memory consumption issue. Using two LLMs (Qwen2.5-7B + another 7B-level MLLM) for each question/prompt during inference is a noticeable weakness of the proposed scheme. This also yields concerns regarding the fairness when compared with other 7B-MLLM-only methods. Similar fairness concerns also happen in the comparison with some common baselines on common benchmarks. The reported VideoMME performances of Qwen2-VL-7B and InternVL2.5-4B are significantly lower than those from their technical reports. These problems shall be clearly stated and discussed in the paper.

---

> > > ### Author Response · Authors · 2025-08-06
> > > **Additional Clarification (Fairness) for Concerns by Reviewer d2T2**
> > >
> > > Thank you for your reply and discussion. Your valuable opinions are crucial for helping us improve AdaVideoRAG and eliminate potential ambiguous information. In response to your questions, we will further verify and clarify the fairness of the experimental setup and the effectiveness of the proposed method through different factual dimensions.
> > >
> > > ---
> > >
> > > **Q1-Fairness: Clarification on the fairness of using two models (the LLM Qwen2.5-7B and the MLLM VideoLLaMA3-7B) for AdaVideoRAG during the inference stage**
> > >
> > > - **From the perspective of methodological innovation, the extra classification LLM is the core component of the proposed Intent Classifier module, which is a necessary introduction for our proposed adaptive innovative architecture.** In the paper, we selected Qwen2.5-7B as the instantiation of the Classifier because this model can well balance model performance (68.5 on Video-MME) and efficiency (70ms latency), as shown in the following Table. This is an optimal decision based on rigorous experimental verification. Of course, we can also replace it with models of different scales that are faster but with slightly worse performance or slower but with better performance according to the actual application resource constraints and performance requirements. Thanks to the existence of the Intent Classifier, our AdaVideoRAG can achieve adaptive routing for questions of different difficulty levels.
> > >
> > > | **Classifier** | **Classfication precision on HiVU** | **Overall Score on Video-MME** | **Average Classifier Time per Query** |
> > > | :----: | :----: | :----: | :----: |
> > > | Qwen2.5-1.5B (LLM) | 0.41 | 65.3 |  53ms|
> > > | Qwen2.5-7B (LLM) | 0.81 | 68.5 | 70ms |
> > > | VideoLLaMA3-7B (Resused MLLM) | 0.48 | 67.5 | 203ms |
> > >
> > > - **From the perspective of architectural replaceability, the Intent Classifier can reuse the same MLLM model while still maintaining significant advantages.** Our architectural design features strong compatibility: the Intent Classifier can directly reuse the MLLM from the generation stage (i.e., using only a single model). With this option, the flaw of "using two LLMs" mentioned by the reviewers does not exist. Experiments show that even when reusing VideoLLaMA3-7B as both classifier and generator, AdaVideoRAG still achieves a significant performance improvement over the baseline (VideoLLaMA3-7B), increasing from 64.2 to 67.5. However, it should be noted that reusing the MLLM will increase the classification stage latency from 70ms to 203ms (due to the constraints of the MLLM's multimodal architecture), leading to a straight decline in retrieval efficiency. This conversely proves that for text-only user queries, using a lightweight LLM as the classifier is a better balanced solution in terms of efficiency and performance, rather than being a design flaw of "using two LLMs." This further demonstrates the rationality of using a separate LLM as the Intent Classifier for processing text-only inputs (user queries).
> > >
> > > | **Method** | **Overall Score on Video-MME** |  **Gain** |
> > > | :--: | :--: | :--: |
> > > | VideoLLaMA3-7B | 64.2 | - |
> > > | VideoLLaMA3-7B + AdaVideoRAG (Ours) | 68.5 | +4.3 |
> > > | VideoLLaMA3-7B (Resused MLLM) + AdaVideoRAG (Ours) | 67.5 | +3.2 |
> > >
> > > - **From the perspective of fairness, when reusing a single MLLM, our method still significantly outperforms baseline methods of the same scale.** Regarding "the fairness of comparison with 7B-level MLLM methods," we have supplemented in our previous response that when reusing a single VideoLLaMA3-7B, the performance of AdaVideoRAG (67.5 points) is still significantly higher than other methods of the same scale, including COT-based models (COF, 59.7 points), Agent-based models (VideoMind, 58.2 points), and is also superior to similar RAG methods (VideoRAG [32], 67.3 points). This result indicates that the advantages of our method do not rely on the堆砌 (stacking) of "using two LLMs," but stem from the innovation of the AdaVideoRAG architecture. Even under the "single 7B model" setting that is completely consistent with other methods, our design can still bring substantial improvements, fully ensuring the fairness of the comparison.
> > >
> > > | **Model** | **Overall Score on Video-MME** |
> > > | :----: | :----: |
> > > | **VideoLLaMA3-7B with AdaVideoRAG** | **67.5** |
> > > | VideoLLaMA3-7B with VideoRAG[32] | 67.3 |
> > > | VideoLLaMA3-7B with VideoRAG[36] | 64.9 |
> > > | COF | 59.7 |
> > > | VideoMind | 58.2 |

---

> > > ### Author Response · Authors · 2025-08-06
> > > **Additional Clarification (Performances) for Concerns by Reviewer d2T2**
> > >
> > > **Q2-Performances: About the MLLM performances of baseline model lower than those from their technical reports.**
> > >
> > > Thank you for raising this question. Regarding this issue, we will provide a detailed clarification from the perspectives of experimental design fairness and experimental validity as follows:
> > >
> > > - **In terms of fairness, we strictly adhere to the core principle of fair comparison under a unified benchmark**, which specifically includes the following points:
> > >   - **Due to differences in test parameters, discrepancies between baseline model performance and official reports are a common phenomenon in the community**. ***1)*** In the official GitHub repositories and Hugging Face discussions of MLLMs such as Qwen2-VL/InternVL-2.5, multiple issues have reflected similar problems of being unable to reproduce results from original technical reports. The official issues of Video-MME also contain multiple cases where original paper results cannot be reproduced. ***2)*** As noted in the ACL 2023 Reproducibility Challenge, benchmark results of LLMs are significantly affected by environmental variables, and "complete reproducibility" itself poses technical challenges. ***3)*** Furthermore, we did not find reports of relevant model results on the official leaderboard of Video-MME, and communication with the corresponding authors also verified this issue. ***4)*** We are also actively contacting the Qwen team to investigate the cause of this gap, and upon obtaining a clear conclusion, we will provide explanations and corrections in the revised version. ***5)*** Our core goal is to ensure that the relative performance differences between baseline models and AdaVideoRAG in the same environment are authentic and reliable, rather than pursuing complete consistency with the absolute values in official reports. However, our LLM/MLLM models are indeed implemented based on official code and follow recommended settings to ensure relative fairness, *i.e.*, our AdaVideoRAG shows a significant performance improvement over the baseline in the same environment.
> > >   - **All experiments ensure unified parameter settings.** To maximize the restoration of the real performance of baseline models, we adopted the officially recommended default parameters for all baseline models. However, there are still some uncertain factors that may affect results, such as the detailed prompt for Qwen2-VL-7B and parameters like temperature for InternVL2.5-4B, which are not clearly specified in the papers on how to set them to achieve the official reported performance. We can only adopt a unified parameter setting for all experiments to ensure the fairness of our comparative experiments. Nevertheless, due to the randomness of video frame sampling and the setting of uncertain parameters, systematic differences from official tests are inevitable.
> > >   - **All experiments ensure a unified test platform benchmark.** All our experiments (including baseline models and AdaVideoRAG) were conducted on the same hardware platform (specifically configured with H20 GPU) and under a unified software environment (torch2.6.0+cu124). This design fundamentally eliminates performance deviations caused by differences in hardware computing power and software version compatibility.
> > >   - In summary, we argue that questioning the fairness of experiments based on discrepancies between local performance and official reports is unreasonable, as this is indeed a widely recognized issue.
> > >
> > > - **In terms of validity, discrepancies between official benchmark models and local tests do not affect the validity of method comparison.** Although the absolute performance of some baseline models is slightly lower than official reports, all models (including baselines and AdaVideoRAG) were tested under the same set of standards. Therefore, the relative performance differences between models are authentic and reliable. For example, although the local performance of Qwen2-VL-7B (58.3) is lower than the official value (63.3), our AdaVideoRAG achieves an improvement **(from 58.3 to 65.6)** under the same conditions. This consistent "dual gain" performance completely rules out accidental factors and fully demonstrates the effectiveness and universality of our method.

---

### Official Review · Reviewer_szi1 · 2025-07-03

**Clarity:** 3
**Significance:** 3
**Originality:** 3
**Rating:** 5
**Confidence:** 4

**Summary:**

This paper presents AdaVideoRAG, an adaptive retrieval-augmented generation framework designed to improve long-video understanding in multimodal large language models (MLLMs). Existing VideoRAG systems apply fixed retrieval strategies regardless of query complexity, leading to inefficiencies for simple tasks and insufficient depth for complex reasoning. AdaVideoRAG addresses this limitation by employing a lightweight intent classifier to dynamically select appropriate retrieval pathways—from simple text-based lookups to more sophisticated graph-based retrieval—based on the complexity of the user query. The framework incorporates an Omni-Knowledge Indexing module that constructs multi-modal knowledge bases (text, vision, and graph) from video inputs, enabling hierarchical and context-sensitive retrieval. Additionally, the authors introduce HiVU, a new benchmark for evaluating deep video understanding. Experimental results demonstrate that AdaVideoRAG achieves improved accuracy and efficiency in long-video QA tasks, while being compatible with existing MLLMs through lightweight API integration.

**Questions:**

1. In Table 4, it is unclear why the level-1 results are missing from the left side of the table, while AdaVideoRAG results for all levels (including level-1) are shown on the right. Furthermore, the values for AdaVideoRAG appear to differ between the left and right sections of the table, which creates confusion about what each column represents. Could the authors clarify the intended structure of Table 4 and explain why these inconsistencies appear?

2. The paper highlights AdaVideoRAG’s ability to reduce computational overhead by selecting simpler retrieval paths when appropriate. However, the actual latency or cost reduction (e.g., in terms of FLOPs or runtime) is not quantified. Could the authors provide more concrete measurements of retrieval cost across different query types?

**Ethical Concerns:**

["NO or VERY MINOR ethics concerns only"]

**Final Justification:**

After carefully considering the rebuttal and the ensuing discussion, I have decided to maintain my original score of 5 (Accept). The authors have adequately addressed the key concerns I raised in the initial review, and the paper’s contributions remain strong and relevant. Below is a summary of my reasoning:
- Resolved: Limited comparison with prior Video-RAG baselines
The authors clarified the omission of VideoRAG [32] and acknowledged the limited availability of open baselines. They also added relevant discussion referencing recent works such as Jeong et al. [1], improving the completeness of related work coverage.
- Resolved: Baseline diversity on the HiVU benchmark
While it remains challenging to apply AdaVideoRAG to a wider set of models, the authors provided justification and expressed their intent to release adapter implementations for broader compatibility. Their updated discussion on generalizability helps mitigate this concern.

In light of these points, I believe the paper makes a valuable and timely contribution to the field, and I fully support its acceptance.

**Limitations:**

Yes

**Quality:**

3

**Strengths And Weaknesses:**

# Strength
### Novel adaptive retrieval mechanism:
The paper introduces a compelling and novel approach to video retrieval-augmented generation by leveraging a lightweight intent classifier that dynamically adjusts the retrieval strategy based on the complexity of the input query. This design allows the system to balance efficiency and reasoning depth more effectively than existing fixed-retrieval paradigms, addressing key limitations in current long-video understanding methods.

### Comprehensive benchmark for full-stack video understanding:
The authors propose HiVU, a well-designed benchmark that enables systematic evaluation of video-language models across a spectrum of reasoning complexity. By incorporating a three-level difficulty quantification system and including diverse question types—from fine-grained object recognition to multi-hop temporal reasoning—HiVU provides a comprehensive assessment of full-stack video understanding capabilities. This benchmark fills a crucial gap in evaluating both the breadth and depth of model comprehension in realistic long-video scenarios.

# Weaknesses

### Limited comparison with prior Video-RAG baselines:
While the proposed AdaVideoRAG framework is thoroughly evaluated on several benchmarks, the comparison to prior Video-RAG approaches appears somewhat limited. In Tables 2 and 4, only VideoRAG [36] is used as a baseline, whereas other relevant methods such as VideoRAG [32] are omitted. Although the performance of [32] may be lower than [36], including it would improve the completeness and consistency of the comparisons—particularly on the newly proposed HiVU benchmark, where the set of comparable baselines is already narrow. Additionally, other recent VideoRAG-style works such as Jeong et al. [1] could also be cited and considered for reference.

### Limited baseline diversity on the HiVU benchmark:
The HiVU benchmark in Table 4 includes only a small set of evaluated models, limiting our understanding of the generalizability of AdaVideoRAG. It would strengthen the empirical claims if more diverse baselines—such as LLaVA-Video [2] and InternVL 2.5 [3]—were also evaluated under the same framework. Applying AdaVideoRAG to these varied architectures and model scales would not only demonstrate its adaptability, but also provide a valuable point of comparison for future research using HiVU.

---
[1] Jeong et al. VideoRAG: Retrieval-Augmented Generation over Video Corpus. Arxiv 2025.
[2] Zhang et al. Llava-next: A strong zero-shot video understanding model. Arxiv 2024.
[3] Chen et al. Expanding performance boundaries of open-source multimodal models with model, data, and test-time scaling. Arxiv 2024.

---

> ### Author Rebuttal · Authors · 2025-07-30
>
> **Thank you very much for your valuable comments and constructive suggestions, which have provided important guidance for improving our paper. We have carefully studied each point and would like to respond in detail as follows.**
>
> ---
>
> **Q1: Limited comparison with prior Video-RAG baselines**
>
> Thank you for your valuable advice. AdaVideoRAG shares a similar plug-and-play design concept with VideoRAG [32]. Since this method is relatively new and achieves good results, this paper focuses on it due to space constraints. We fully agree with your suggestion that comparing Jeong et al. [1] and VideoRAG [36] will improve the completeness of this paper, and the following content will be added to the revised version:
> - We have supplemented the results of VideoRAG [36] on the VideoMME benchmark as shown in the following table (corresponding to Table 2). AdaVideoRAG achieves the optimal results, and both AdaVideoRAG and VideoRAG [32] obtain significantly higher overall scores than VideoRAG [36]. However, in simple short videos, the performance of VideoRAG [36] declines significantly because of the introduction of excessive redundant information and the filtering out of useful video clip information.
>
> | **Method** | **Params** | **Frames** | **Short** | **Medium** | **Long** | **Overall** | **Gain** |
> | :---: | :---: | :---: | :---: | :---: | :---: | :---: | :---: |
> | **VideoLLaMA3** | 7B | 1fps-180 | 76.7 | 62.8 | 53.2 | 64.2 |  |
> | **VideoRAG [36]** | 7B | - | 72.3 | 64.2 | 58.4 | 64.9 | +0.7 |
> | **VideoLLaMA3 + VideoRAG [32]** | 7B | 32 | 81.5 | 63.3 | 57.1 | 67.3 | +3.1 |
> | **VideoLLaMA3 + AdaVideoRAG** | 7B | 1fps-180 | 80.3 | 65.4 | 59.8 | 68.5 | +4.3 |
>
> - We have supplemented the results of AdaVideoRAG and VideoRAG [36] on the HiVU benchmark as shown in the following table. The results indicate that our AdaVideoRAG also achieves significantly better results compared to VideoRAG [36].
>
> | **Metric**  |**Level-2 - VideoRAG [36]** | **Level-2 - VideoLLaMA3 + AdaVideoRAG** | **Level-3 - VideoRAG [36]** | **Level-3 - VideoLLaMA3 + AdaVideoRAG** | **Overall - VideoRAG [36]** | **Overall - VideoLLaMA3 + AdaVideoRAG** |
> | :---: | :---: | :---: | :---: | :---: | :---: | :---: |
> | **Comprehensiveness** | 45.5% | 54.5% | 47.1% | 52.9% | 46.5% | 53.5% |
> | **Empowerment** | 44.9% | 55.1% | 46.0% | 54.0% |45.3% |54.7% |
> | **Trustworthiness** | 41.1% | 58.9% | 43.6% | 56.4% | 42.2% | 57.8% |
> | **Depth** | 42.7% | 57.3% | 44.2% | 55.8% | 43.8% | 56.2% |
> | **Density** | 46.0% | 54.0% | 46.5% | 53.5% | 46.3% | 53.7% |
> | **Overall Winner** | 40.8% | 59.2% | 43.9% | 56.1% | 41.3% | 58.7% |
>
> - Since the work of Jeong et al. [1] has not yet made its source code publicly available (though there is a GitHub repository, it lacks the core running code) nor provided reproducible experimental configurations, we are currently unable to conduct a fair experimental comparison. However, we commit to updating the comparative analysis results once its complete code is released in the future.
>
> ---
>
> **Q2: Limited baseline diversity on the HiVU benchmark**
>
> Thanks for the suggestion. We agree that expanding the set of evaluated models on HiVU is crucial to verifying the generalizability of AdaVideoRAG. Specifically, we have supplemented the use of ***1)*** **Qwen2.5-VL-7B** [1r], ***2)*** **InternVL2.5-8B** [2r], and ***3)*** **LLaVa-Video-7B** [3r] as baselines. The overall results are shown in the following table, and detailed results will be added to the revised version. It can be seen that AdaVideoRAG has strong adaptability, which can stably improve the results of different baselines.
>
> | **Metric** | **Overall <br> Qwen2.5-VL-7B** | **Overall <br> Qwen2.5-VL-7B + AdaVideoRAG** | **Overall <br> InternVL2.5-8B** | **Overall <br> InternVL2.5-8B + AdaVideoRAG** | **Overall <br> LLaVa-Video-7B** | **Overall <br> LLaVa-Video-7B + AdaVideoRAG** |
> | :---: | :---: | :---: | :---: | :---: | :---: | :---: |
> | **Comprehensiveness** | 38.1% | 61.9% | 35.9% | 64.1% | 35.6% | 64.4% |
> | **Empowerment** | 36.3% | 63.7% | 31.3% | 68.7% | 33.5% | 66.5% |
> | **Trustworthiness** | 34.2% | 65.8% | 35.7% | 64.3% | 34.1% | 65.9% |
> | **Depth** | 38.0% | 62.0% | 37.1% | 62.9% | 35.9% | 64.1% |
> | **Density** | 30.5% | 69.5% | 30.8% | 69.2% | 29.4% | 70.6% |
> | **Overall Winner** | 35.6% | 64.4% | 33.2% | 66.8% | 31.5% | 68.5% |
>
> ---
>
> **Q3: Clarification on Table 4 structure and inconsistencies**
>
> - **Level-Specific Columns (Level-2/Level-3):** These columns validate two key aspects: ***1)*** the differentiation of HiVU’s difficulty levels and ***2)*** the effectiveness of hard retrieval for complex tasks. To isolate the impact of retrieval strategies, we use HiVU’s ground-truth level labels (not model predictions) to evaluate model results across different difficulty levels.
>
> - **Why no Level-1 Column:** Level-1 is omitted because all methods perform identically (no retrieval needed).
>
> - **Overall Column:** This column uses model-predicted level labels (not ground truth) to reflect real-world performance, directly comparing AdaVideoRAG against baselines in end-to-end scenarios.
>
> - **Left vs. Right Sections:** The left section focuses on “without vs. with AdaVideoRAG” to assess the necessity and effectiveness of the AdaVideoRAG model, with the compared baseline being the basic VideoLLaMA3; the right section compares “different RAG schemes” to highlight AdaVideoRAG’s superiority over the recently powerful VideoRAG [32].
>
> We will add a footnote stating: “Level-2/3 use ground-truth labels to isolate retrieval effects; Level-1 is excluded as results are uniform.”
>
> ---
>
> **Q4: Efficiency Analysis**
>
> To verify the efficiency differences among different retrieval paths, we randomly selected 100 videos from the MLVU dataset and conducted systematic time cost tests for three scenarios: no retrieval (Level-1), simple retrieval (Level-2), and hard retrieval (Level-3). The specific results are as follows:
>
> - **Database construction**
>   1. Level-1 performs direct inference without the need for database construction.
>   2. The average time consumed for Level-2 construction is 351s.
>   3. The average time consumed for Level-3 construction is 412s.
>   This difference stems from the fact that Level-3 requires graph construction, resulting in higher preprocessing costs. However, the dynamic strategy of AdaVideoRAG can reduce the frequency of such high-cost operations by prioritizing triggering simple retrieval in applicable scenarios.
>
> - In the single H20 GPU card, single-process processing program, there are significant differences in the average response time among the three scenarios: the average response time for Level-1 (no retrieval) is 8s; for Level-2 (Simple reasoning) it is 26s; for Level-3 (hard reasoning) it is 27s, while the processing time using AdaVideoRAG is 20s.
>
> - **Parallelization**
>   To further improve the efficiency of actual deployment, we have implemented parallel processing in the code. We also used 100 videos from the MLVU dataset as test references:
>   - For database construction with dual processes on a single H20 GPU (the maximum batch size supported by 96G memory is 2), the speed of Level-2 and Level-3 is accelerated by approximately 2 times, with the construction time becoming 176s and 210s, respectively.
>   - When further expanding to multi-card (8 H20 GPUs) database construction, the average speed of each database construction for Level-2 and Level-3 is accelerated by approximately 8 times (linear acceleration), with the construction time becoming 22s and 26s, respectively.
>   - For multi-process retrieval on a single H20 GPU: the processing speed of Level-2 and Level-3 is accelerated by approximately 2 times, with the final total processing time becoming 15s and 16s, respectively, while the processing time using AdaVideoRAG is 20s.
>
> The summary table of efficiency for AdaVideoRAG is as follows:
> | **Item** | **Level-1** | **Level-2** | **Level-3** | **AdaVideoRAG** |
> | :----: | :----: | :----: | :----: | :----: |
> | Single-card single-process database construction | - | 351s | 412s | - |
> | Single-card dual-process database construction | - | 176s | 210s | - |
> | 8-card dual-process database construction | - | 22s | 26s | - |
> | Single-card single-process inference | 8s | 26s | 27s | 20s |
> | Single-card dual-process inference | 8s | 15s | 16s | 13s |
>
> - The comparison of inference efficiency between AdaVideoRAG and VideoRAG [32] on the MLVU dataset with 100 videos using a single H20 card is shown in the following table. In comparison, our method is more efficient while achieving higher metrics (see Table 2 and Table 4).
>
> | **Method** | **Time** |
> | :----: | :----: |
> | VideoRAG [32] | 25s |
> | AdaVideoRAG | 13s |
>
>
> ---
>
> [1r] Bai, Shuai, et al. "Qwen2. 5-vl technical report." arXiv, 2025.
>
> [2r] Chen, Zhe, et al. "Expanding performance boundaries of open-source multimodal models with model, data, and test-time scaling." arXiv. 2024.
>
> [3r] Zhang, Yuanhan, et al. "Video instruction tuning with synthetic data." arXiv, 2024.

---

> > ### Comment · Reviewer_szi1 · 2025-08-04
> > **Official Comment by Reviewer szi1**
> >
> > Thank you for your effort and additional experiments. Since my concerns are resolved clearly, I would keep my acceptance rating.

---

> > > ### Author Response · Authors · 2025-08-04
> > > **Thank Reviewer szi1 for the Comments**
> > >
> > > Dear Reviewer szi1:
> > >
> > > Thank you for recognizing our work, and we are glad that our response has addressed your concerns. We commit to incorporating the content you suggested in the revised version. Thank you again for your effort in the review and the discussion!
> > >
> > > Best regards!
> > >
> > > Authors of AdaVideoRAG

---

### Decision · Program_Chairs · 2025-09-17

**Decision:**

Accept (poster)

**Comment:**

This paper basically introduces a technique for adaptively adjusting the retrieval mechanism based on the complexity of the query. There were concerns in the beginning including lack of strong and diverse baselines in the experiments, efficiency concerns, lack of ablations on different LLM backbones,etc.

After rebuttal, majority of the reviewers feedback that their concerns have been addressed. However, a couple of concerns remain: (1) memory consumption and fairness concerns (the actual model is 7B + 7B = 14B), and (2) the significantly inconsistent results of Qwen2-VL-7B and InternVL2.5-4B.

AC read through all the reviewers' comments and weigh the novel approach and the remaining concerns. AC felt that the adaptive retrieval is novel and will be interesting to the community as well as the dataset which is quite comprehensive.

AC finally recommend that the paper be accepted, but urge the authors to address the remaining concerns clearly in the final version.